# LogicSR: A Unified Benchmark for Logical Discovery from Data

## Abstract

Discovering underlying logical expressions from data is a critical task for interpretable AI and scientific discovery, yet it remains poorly served by existing research infrastructure. The field of Symbolic Regression (SR) primarily focuses on continuous mathematical functions, while Logic Synthesis (LS) is designed for exact, noise-free specifications, not for learning from incomplete or noisy data. This leaves a crucial gap for evaluating algorithms that can learn generalizable logical rules in realistic scenarios. To address this, we introduce LogicSR, a large-scale and comprehensive benchmark for logical symbolic regression. LogicSR is built from two sources: real-world problems from digital circuits and biological networks, and a novel synthetic data generator capable of producing a diverse set of complex logical formulas at scale. We use LogicSR to conduct a rigorous evaluation of 17 algorithms, spanning classical logic solvers, modern machine learning models, and Large Language Models (LLMs). Our findings reveal that the logical modeling capabilities and generalization robustness of these algorithms significantly depend on task scale and logical complexity, with current cutting-edge LLMs showing limited complex logical reasoning ability. LogicSR provides a robust foundation to benchmark progress, unify evaluation across disparate fields, and steer the future development of powerful neuro-symbolic systems.

## 1 Introduction

The discovery of underlying logical expressions from data, which we refer to as *logical symbolic regression*, is a key challenge in artificial intelligence. Uncovering these expressions is highly valuable. It can provide human-interpretable rules for complex phenomena (Marques-Silva & Ignatiev, 2022) and generate programs that can be directly synthesized into electronic circuits (Nane et al., 2015). These practical applications make logical symbolic regression a crucial tool for advancing interpretable modeling and automated scientific discovery in domains governed by discrete logic, such as digital systems and gene regulatory networks.

Despite its importance, logical symbolic regression is underexplored by current research communities. Specifically, it falls into a research gap between two established fields, **Symbolic Regression (SR)** and **Logic Synthesis (LS)**. SR traditionally uses methods like genetic programming, and has experienced a resurgence with new machine learning-based approaches (Mundhenk et al., 2021; d'Ascoli et al., 2023). However, SR research and its associated benchmarks have primarily focused on recovering continuous mathematical formulas involving operators such as addition, multiplication, and transcendental functions. Meanwhile, the expressions studied in SR are usually limited to small scales (e.g. less than 10 variables and operators). There is a distinct lack of standardized benchmarks for discovering complex expressions in the Boolean domain.

On the other hand, the field of Logic Synthesis (LS), central to Electronic Design Automation (EDA), excels at manipulating and optimizing logical expressions for hardware design (Brayton & Mishchenko, 2010; Chowdhury et al., 2023). However, LS methods are mostly based on symbolic search techniques engineered for scenarios with exact and complete specifications (e.g., a full truth table). They are not designed to find approximate solutions from noisy or incomplete data, nor do they prioritize generalization to unseen inputs—properties that are paramount for machine learning applications. This leaves a critical gap for a benchmark designed to evaluate algorithms on their ability to learn approximate and generalizable logical rules from data.

While SR and LS share a similar objective, their respective communities have evolved in isolation. Emerging techniques like large language models (LLMs) also exhibit a certain ability in generating mathematical formulas from data samples, yet their performance relative to traditional algorithms remains unknown. A unified benchmark is therefore essential to rigorously compare these disparate approaches and foster cross-domain innovation.

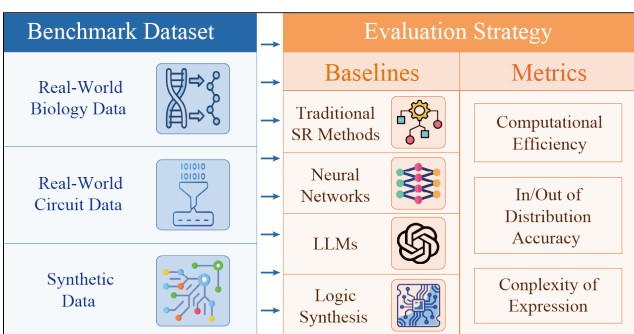

Figure 1: Main components of the LogicSR benchmark.

As shown in Table 1, existing benchmarks each exhibit clear limitations. To fill this void, we introduce LogicSR, a large-scale, multi-granularity benchmark for logical symbolic regression. Our benchmark is based on two primary sources. First, we curate a real-world dataset from application domains where logical reasoning is fundamental, including data-driven logic synthesis and boolean models of biological gene regulatory networks (Kadelka et al., 2024). Second, we propose a scalable synthetic generation algorithm to generate a large set of random logical expres-sions with different scales. This algorithm leverages truth table analysis, structured sampling, and graph-based composition to produce a diverse set of complex, non-redundant logical formulas. We further add different levels of noise and randomly sample the training data points for large expres-sions, closely mimicking practical scenarios with noisy and incomplete data. The main components of LogicSR are shown in Figure 1.

Using our benchmark, we conduct a fine-grained evaluation of 17 representative algorithms, cov-ering classical logic-based methods, modern machine-learning models, and large language models. Overall, the results show that existing approaches struggle with moderately scaled logical discov-ery tasks, either requiring substantial computational resources or delivering unsatisfactory accuracy. More specifically: (1) traditional logic synthesis performs well on small-scale problems but lacks scalability and flexibility; (2) ML and symbolic-regression models generalize better but come with significant computational overhead; and (3) current LLMs still show limited capacity for handling complex logical reasoning. These findings collectively reveal challenges in scalability, noise robust-ness, and operator-set compatibility across current methods.

In summary, our main contributions are as follows.

- **A new benchmark for logical symbolic regression**, featuring a large and diverse collection of problems derived from both real-world applications and a novel, high-quality synthetic gen-eration process.

- **An extensive, cross-domain evaluation** of 17 diverse algorithms, evaluating performance on metrics including accuracy in distribution, generalization outside distribution, simplicity of expression and computational speed.

- **A foundational set of findings and analyses** that map the capability boundaries of current methods, and providing insights to guide further research towards addressing existing limita-tions in logical reasoning tasks.

We hope this study offers a solid foundation for evaluating and improving algorithms in logical discovery, bridging the gap between theoretical symbolic models and practical reasoning.

## 2 PROBLEM DEFINITION

The goal of logical symbolic regression is to identify a logical formula from a set of input-output examples. An ideal benchmark must provide datasets that are formally sound while also being diverse, scalable, and representative of real-world challenges like noise and incomplete information. We formalize the task and the structure of the target expressions as follows.

Table 1: Comparison of existing logic and symbolic regression benchmarks.

| Benchmark / Feature | Domain | Scalability | #Samples | Noise / Sampling | MIMO | Boolean | Metrics |
|---|---|---|---|---|---|---|---|
| EPFL (Amarú et al., 2015) | Circuit | Fixed | 23 | $\times$ | $\times$ | $\checkmark$ | Size / Depth |
| ISCAS'85/'89 (Brglez et al., 1989) | Circuit | Fixed | 85: 10, 89: 31 | Partial | $\checkmark$ | $\checkmark$ | Size / Depth |
| IWLS'25 (IWLS, 2025) | Circuit | Fixed | 100 | Partial | $\times$ | $\times$ | Size / Depth |
| LLM-SRBench (Shojaee et al., 2025) | Real-valued | Scalable | 239 | $\checkmark$ | $\times$ | $\times$ | NMSE, numeric precision, etc |
| SRBench (Cava et al., 2021) | Real-valued | Fixed | 252 | $\checkmark$ | $\times$ | $\checkmark$ | Complexity, $R^2$ |
| BBM Boolean Models (Pastva et al., 2023) | Biological networks | Fixed | 245 | Partial | $\checkmark$ | $\checkmark$ | Function-level metrics |
| MCBF (Subbaroyan et al., 2022) | Biological networks | Fixed | 2687 | $\checkmark$ | $\checkmark$ | $\checkmark$ | Boolean complexity, sensitivity, enrichment $p$-value |
| **LogicSR (ours)** | **Boolean networks** | **Scalable (5–80+)** | **1000+** | $\checkmark$ | $\checkmark$ | $\checkmark$ | **Bit/Sample accuracy, complexity** |

**Definition 1** (Logical Symbolic Regression Task). *Let $f : \{0,1\}^k \to \{0,1\}$ be an unknown Boolean function. Let $\mathcal{D} = \{(x^{(i)}, y^{(i)})\}_{i=1}^N$ be a dataset of $N$ input-output pairs, where $x^{(i)} \in \{0,1\}^k$, $y^{(i)} \in \{0,1\}$, and each pair is a (possibly noisy) sample from $f$. The task is to find a logical expression, represented as a logic network $\mathcal{G}'$, that implements a function $f'$ which best approximates $f$ on both seen and unseen data.*

**Definition 2** (Logic Network). *A logic network is a tuple $\mathcal{G} = (\mathcal{V}, \mathcal{E}, \mathcal{I}, \mathcal{O})$ where: $\mathcal{V}$ is the set of nodes, consisting of input nodes and internal logic–gate nodes. Each internal node applies a Boolean operator from a predefined operator set $\mathcal{U}$ (e.g., $\{AND, OR\}$). $\mathcal{E} \subseteq \mathcal{V} \times \mathcal{V}$ is the set of directed edges with an optional inversion; $\mathcal{I} \subseteq \mathcal{V}$ is the set of input nodes, with an in-degree of zero; $\mathcal{O} \subseteq \mathcal{V}$ is the set of nodes chosen as output, with an out-degree of zero. A logic network $\mathcal{G}$ should satisfy the following conditions: (a) **Connectivity:** Every non-input node $v \in \mathcal{V} \setminus \mathcal{I}$ must be reachable from at least one input node $v_i \in \mathcal{I}$ (i.e. not isolated). (b) **Acyclicity:** The graph $\mathcal{G}$ must be a Directed Acyclic Graph (DAG) to prevent trivial logic.*

In essence, **Definition 1** specifies the learning problem and its solution space, while **Definition 2** describes the structural form of candidate solutions. The overall objective is to obtain a solution with maximal accuracy and minimal complexity.

## 3 THE LOGICSR BENCHMARK

LogicSR provides a diverse suite of problems by 1) curating real-world datasets, 2) proposing a scalable synthetic data generator, 3) incorporating noise model with controllable noise injection, and 4) using incomplete data samples to test generalization.

### 3.1 REAL-WORLD DATASETS

**Circuit Dataset.** This dataset is derived from the International Workshop on Logic & Synthesis (IWLS) benchmarks, which represent challenges in real-world circuit design (IWLS, 2025). These problems are characterized by high structural complexity, significant fan-out, reconverging paths, and multi-output functions, making them a practical testbed for symbolic reasoning. We processed the original IWLS benchmarks, which use an And-Inverter Graph (AIG) format, to generate explicit truth tables. The original format often assumes an implicit ordering of input variables, which is suitable for specialized LS tools like ABC (Brayton & Mishchenko, 2010) but not for general SR methods. We converted the data into an explicit format, making it accessible to a wider range of algorithms.

**Biological Dataset.** The BioDivine Boolean Models (BBM) dataset contains 245 Boolean network models of real biological systems, such as gene regulatory networks (Pastva et al., 2023). These

networks capture the complex, nonlinear logic inherent in biology. To create benchmark tasks, we treated each node in a given biological network as a distinct Boolean function. For each function, we extracted its corresponding combinational logic, where the inputs are the states of its regulatory genes and the output is its own updated state. This process yields a rich collection of naturally-occurring, meaningful Boolean functions.

## 3.2 SYNTHETIC DATASET GENERATION

While real-world datasets provide relevance, they are limited in scale and diversity. To overcome this, we developed a novel two-stage efficient synthesis process for generating large-scale, complex, and structurally diverse *ground-truth* logic networks from scratch. This approach bypasses the limitations of existing methods that rely on pre-computed libraries or computationally expensive truth-table evaluations.

### 3.2.1 STAGE 1: SMALL-SCALE NETWORK CONSTRUCTION

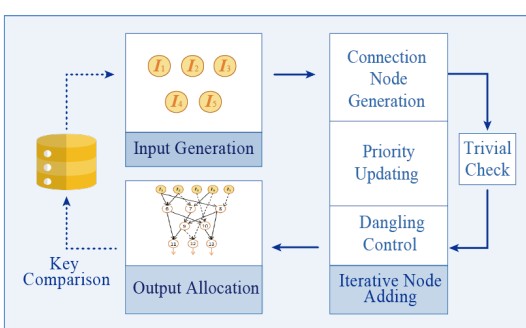

Figure 2: The procedure of small-scale logic network generation.

Each small network $\mathcal{G}_j(K_j, L_j, M_j)$ is generated by iteratively connecting newly created gate nodes $v$ for $M_j$ steps. $K_j$ input nodes are first initialized with truth tables of length $2^{K_j}$ and a priority $p_0$ and added to $\mathcal{G}_j$. While $L_j$ denotes the desired number of outputs for this network.

The iterative process encompasses generating connection nodes, updating priorities, controlling dangling nodes, and verifying triviality. Dangling nodes are those that do not exist on any end-to-end path connecting the network inputs to the outputs. In contrast, trivial nodes are characterized by outputs that convey minimal information, exemplified by truth tables that consistently yield 0 or 1.

During the generation process, candidate pools are initialized with all available nodes, or forced to dangling nodes if their count exceeds output size $L_j$. The algorithm first samples a parent $c_0$ from the pool weighted by priority. The second parent $c_1$ is sampled similarly (since we only involve binary connectives, gate only has two parent nodes). The new node's truth table is computed from its parents with random edge inversions. If the resulting logic is trivial, both parents and inversions are resampled until nontrivial or a trial limit is reached. The process loop for $M_j$ steps, with newly created nodes assigned layer-aware priority:

$$\text{priority}(v) = p_0 + \sqrt{\text{layer}(v)/p_0} \tag{1}$$

The focus on node layers arises from our inclination to link new nodes to deeper network paths, which align with more semantically rich expressions, while parent priorities are diminished by $p_0$ to promote diversity and foster natural complexity growth. Ultimately, outputs are allocated by a preference sequence ⟨ nontrivial-dangling, nontrivial-non-dangling, trivial-dangling, trivial-non-dangling ⟩, with the latter categories being utilized only when the former ones are inadequate. In each set, outputs are sampled using a layer-enhanced priority weight, and a random inversion is applied to each output to promote diversity. The logic of non-dangling nodes is consistently encapsulated within dangling nodes, thereby reducing logical redundancy.

**Key Comparison.** We also adopted a key comparison strategy following (Zhu et al., 2023), where each truth vector is treated as a binary value (taking the inversion vector if its value is smaller), and the network concatenates the sorted truth vectors of all outputs as the key. If two key contradicts, we preserve the one with more layers, as they usually represent greater complexity.

The generation process of the small-scale logic network is illustrated in Figure 2, and the pseudocode is provided in Appendix A.2.

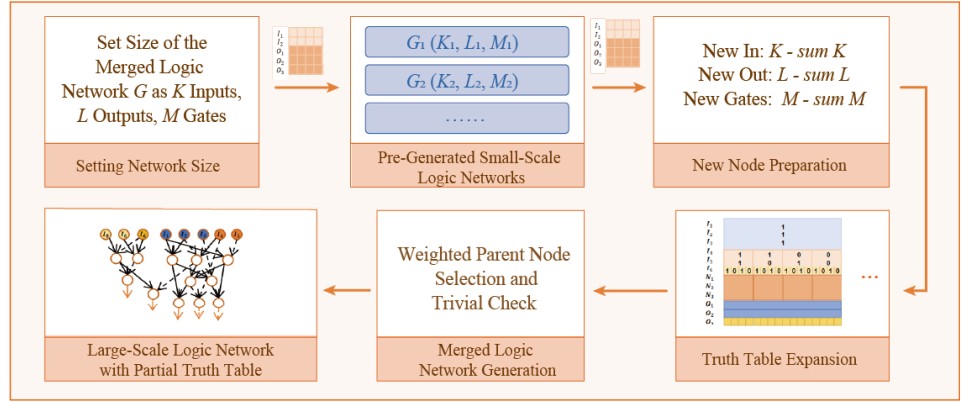

Figure 3: The procedure of large-scale logic network generation. Stage 1 (Small-Scale Generation) produces the building blocks used in large-scale Synthesis, which merges them into complex networks.

### 3.2.2 STAGE 2: LARGE-SCALE NETWORK SYNTHESIS

Large networks $\mathcal{G}(K, L, M)$ are constructed by merging pre-generated sub-networks $\{\mathcal{G}_1, \cdots, \mathcal{G}_\eta\}$ of arbitrary sizes and amounts, provided the sum of $(K_j, L_j, M_j)$ does not exceed the large network's capacity respectively. In addition, the truth tables of each $\mathcal{G}_j$ are expanded to a size of $(K_j + L_j) \cdot T$ with $T < 2^L$, rather than enumerating all possible input combinations. These expanded tables are then concatenated according to their input combinations, a process we refer to as truth-table expansion. The core procedure is as follows: During merged network generation, new inputs are initialized with a priority $p_0$ and a partial truth vetor of length $T$ together covering $T$ combinations.

**Weighted Parent Node Selection and Trivial Check.** For each new gate, parent nodes are sampled by alternating among three modes based on sub-network usage counts: sampling both from the pool of new nodes by priority, one from existing networks and one from the new node pool, or both from two underutilized sub-networks. Inside chosen sub-network(s), node is sampled by weight equation:

$$w(v) = \left(\text{priority}(v) + \sqrt{|\text{Anc}(v)|/p_0}\right)^\alpha \tag{2}$$

where $\alpha > 1$ is an amplification factor, and $\text{Anc}(v)$ represents $v$'s ancestor set. Nodes with a trivial truth table are set zero weight (i.e., $w(v) = 0$), leading progressive complexity growth and network coverage. Finally, after iterative construction and truth table propagation, outputs are assigned using the same strategy as in Stage 1.

The process of large-scale network synthesis is illustrated in Figure 3, and the pseudocode is provided in Appendix A.3.

### 3.3 NOISE MODEL

Bit-flip perturbations simulate the systematic errors commonly observed in biological measurements and circuit-level sampling, and are widely used as a standard noise model in these domains (Trajanovski et al., 2015; Baumann, 2005; Krishnan, 2011). In LogicSR, we use the fixed-budget noise model (Natarajan et al., 2013), which allows the noise magnitude at each preset level to be controlled in a precisely and reproducibly. Specifically, for each Boolean function with $k$ input variables, we have: $y_j : \{0,1\}^k \to \{0,1\}$, we randomly select exactly $\lfloor \eta \cdot 2^k \rfloor$ input combinations in its truth table and flip the corresponding output values, where $\eta$ denotes the noise ratio. Let $S_j$ be the selected subset of inputs (with $|S_j| = \lfloor \eta \cdot 2^k \rfloor$). The noisy output function $\tilde{y}_j(x)$ is then defined as:

$$\tilde{y}_j(x) = \begin{cases} y_j(x) \oplus 1, & \text{if } x \in S_j, \\ y_j(x), & \text{otherwise.} \end{cases} \tag{3}$$

Table 2: Statistics of generated boolean functions under AN and ANO operator sets.

| Operator Set | In/Out Scale | FLV | Max_Depth (Norm.) | Avg_Uniqueness | Global_Uniqueness | Max_Repeat |
|---|---|---|---|---|---|---|
| AN | 5 | 0.3060 | 10.4233 (2.1449) | 0.8553 | 0.7433 | 0.0047 |
| | 10 | 0.6226 | 19.5233 (2.8973) | 0.9900 | 0.9883 | 0.0010 |
| | 20 | 0.4719 | 72.9933 (8.3278) | 0.9252 | 0.9112 | 0.0030 |
| | 40 | 0.5063 | 72.1233 (7.4624) | 0.8898 | 0.8592 | 0.0014 |
| | 80 | 0.5936 | 75.1733 (7.0502) | 0.8898 | 0.8811 | 0.0021 |
| ANO | 5 | 0.3173 | 10.2933 (2.1113) | 0.8520 | 0.7507 | 0.0053 |
| | 10 | 0.6298 | 19.3500 (2.8705) | 0.9930 | 0.9900 | 0.0013 |
| | 20 | 0.4704 | 73.4233 (8.3735) | 0.9212 | 0.9048 | 0.0028 |
| | 40 | 0.4989 | 72.1067 (7.4691) | 0.8947 | 0.8568 | 0.0013 |
| | 80 | 0.5926 | 75.5600 (7.0814) | 0.8894 | 0.8815 | 0.0000 |

### 3.4 INCOMPLETE DATA SAMPLING

To emulate realistic settings where only partial observations are available (Chevalier et al., 2025), LogicSR incorporates incomplete truth-table sampling for both real biological networks and large synthetic Boolean functions, with different sampling rules for the two scenarios. For the BBM dataset, if a gene's regulatory function contains fewer than 12 inputs, we construct its complete truth table. However, when the input dimensionality $\leq 12$, we uniformly sample a subset of input combinations from the full input space $\{0, 1\}^k$ and retain only their corresponding outputs to mimic the sparse subset of discrete states in biological networks. For large-scale synthetic Boolean functions, the number of input combinations grows exponentially with the input dimensionality, making the enumeration of all $2^k$ combinations computationally infeasible for large $k$. To avoid this combinatorial explosion, we set a threshold $L$. When $k \leq L$, we generate complete truth tables; when $k > L$, we uniformly sample $2^L$ input combinations from the full input space $\{0, 1\}^k$ to obtain a partially observed truth table.

## 4 EXPERIMENTS

We conduct all experiments on a server equipped with 8 NVIDIA A800-SXM4-80GB and Intel Xeon Platinum 8358 (2.60 GHz, 64 cores, 128 threads). The dataset is split into training and testing subsets (3 : 1 by default) only when the truth table input count exceeds 7, to preserve information integrity for smaller-scale cases. We configure the LogicSR benchmark construction parameters as follows: $L$ is set to 15; The Noise levels are set to 0%, 1%, and 5%; All input nodes are initialized with the same priority value $p_0 = 3$, and the priority decay coefficient $\mu$ also to $p_0$. We choose 17 representative algorithms from 4 catagories, which are listed below:

- **Logic Synthesis Methods:** ABC (Brayton & Mishchenko, 2010), BDD (Lee, 1959), Espresso (Rudell & Sangiovanni-Vincentelli, 1987) and Quine-McCluskey (Quine, 1952; for Symbolic Logic, 1947);

- **Symbolic Regression Methods:** PySR (Cranmer, 2023), DEAP (Fortin et al., 2012), GPlearn (Stephens & Meeker, 2019) and TenGP (Arenas, 2022);

- **Neural Network-Based Methods:** DiffLogic (Petersen et al., 2022), AI4EDA_TNet (W. et al., 2024), NeuraLUT (Cassidy et al., 2025), SatNet(Wang et al., 2019) and Boolformer (d'Ascoli et al., 2023);

- **LLM-Based Methods:** Feeding the input–output samples to large language models (GPT-4o (Achiam et al., 2023) and GPT-o3 (OpenAI, 2024)) and prompting the model to generate expressions that fit the samples. We also incorporate the Program-of-Thought (PoT) reasoning paradigm (Payoungkhamdee et al., 2025) as additional LLM baselines.

We evaluate the performance of these baselines using the LogicSR benchmark, which includes the aforementioned real-world datasets (BBM and IWLS) and synthetic datasets. The synthetic datasets consist of two expression types: ANO (expressions constructed with AND, NOT and OR gates) and AN (expressions constructed with AND and NOT gates).

We have designed the following unified evaluation metrics as follows:

- **Complexity of Generated Network:** This metric is evaluated by counting the total number of logical operators (AND, NOT and OR if applied) in the logic network generated for each dataset, divided by the number of outputs (see Appendix A.7.1 for details).

- **In-Distribution / Out-of-Distribution Accuracy:** Accuracy is evaluated at three granularities: bit-wise accuracy, sample-wise accuracy (an output is correct only if all bits match), and per-output correctness across all output variables (see Appendix A.7.2 for details).

- **Efficiency:** This metric reflects the total processing time required by an algorithm to produce the final expression or output predictions based on a given dataset. For neural network-based methods, this metric is mainly about the model training time (see Appendix A.7.3 for details).

## 4.1 ANALYSIS OF BENCHMARK COMPLEXITY

In this section, we introduce several commonly used evaluation metrics in logic-synthesis and graph-based benchmarks: 1) Fanout Layer Variance (FLV): Measures the variance of fanout counts across layers, capturing the structural dispersion of signals in the network. 2) Max_Depth: The length of the longest path from primary inputs to outputs, reflecting overall circuit complexity. 3) Normalized Depth: The depth scaled by circuit size, computed as $\frac{max\_depth}{\log_2(\text{Size})}$, which enables fair comparison across networks of different scales. 4) Uniqueness Ratio: Computed by comparing all preserved truth vectors, where two vectors are considered distinct if they differ in any bit. *Average uniqueness* is evaluated within each gate level, whereas *global uniqueness* is computed across the entire network. 5) Max_Repeat: The proportion of all generated functions occupied by the most frequently occurring one. All metrics are reported as averages over multiple generated truth tables. Detailed evaluation results are shown in Table 2.

## 4.2 QUANTITATIVE RESULTS ACROSS DATASETS AND SCALES

Table 3 analyzes the baselines from the perspectives of adaptability and scalability across both synthetic and real-world datasets, with the key observations illustrated below.

Traditional logic synthesis methods maintain 100% training-sample accuracy for small-to-medium scales ($\leq 10$ input variables). However, methods such as ABC, Quine-McCluskey, and BDD rely heavily on exact fitting to training data, resulting in poor generalization and explosive computational costs at larger scales due to redundant rules. Thus, these methods are best suited for small-scale ($\leq 5$) discrete logical tasks. Espresso, however, surpasses this limitation, demonstrating excellent accuracy and generalization capability at input scales up to 20; nevertheless, it remains constrained by the representational bottleneck of its original symbolic output, making it challenging to extend to more complex symbolic scenarios.

Symbolic regression and neural network-based methods leverage formulaic representations and model training, respectively, thereby possessing generalization advantages and compatibility with larger-scale truth tables. In detail, while restricted by their ability to handle large-scale data, PySR, SATNet and Boolformer achieve higher accuracy on small-scale datasets. Both DiffLogic and TenGP exhibit low accuracy, which makes them unsuitable for logical symbol regression tasks.

LLM-based methods are severely limited by context length constraints, rendering them incapable of handling large-scale data. However, on small scale, GPT-o3 demonstrates strong performance, generating precise expressions with low redundancy. GPT-4o, in contrast, lagged significantly behind other baselines, revealing fundamental weaknesses in logical reasoning capabilities. PoT multi-step reasoning improves the accuracy of GPT-4o and GPT-o3, but at the cost of producing significantly more complex expressions.

## 4.3 ANALYSIS OF SCALE, ACCURACY, AND ROBUSTNESS

Figure 4 analyzes the performance differences on the synthetic dataset among various logical modeling methods across three dimensions: input scale (a), bit-level versus sample-level accuracy (b), and robustness against noise (c).

Under noise-free (noise=0) conditions illustrated by Figure 4(a), the relationship between input scale and test sample-wise accuracy reveals the logical complexity boundaries for various mod-

Table 3: Performance of different methods on different datasets (without noise). The top row in each cell is the train and test accuracy (no test accuracy in in5_out5 columns because the small-scale networks do not have a test split). The bottom value in parentheses is the complexity. "–" means that the method cannot handle this scale within the given time, while "N/A" means the method was not tested due to operator set incompatibility.

| | | Synthetic ANO | | | BBM | Synthetic AN | | | IWLS |
|---|---|---|---|---|---|---|---|---|---|
| | | in5_out5 | in10_out10 | in20_out20 | (ANO) | in5_out5 | in10_out10 | in20_out20 | (AN) |
| SR | DEAP | 51.1 (7.5) | 5.9 7.1 (9.2) | – | 18.4 18.3 (6.5) | 39.0 (5.8) | 4.4 5.6 (7.8) | – | 5.5 5.5 (17.1) |
| | GPlearn | 36.7 (3.8) | 5.7 6.9 (3.2) | 4.2 4.3 (1.1) | 15.5 15.3 (3.1) | 28.5 (3.3) | 3.7 4.8 (3.0) | 3.3 3.3 (0.9) | 12.5 12.6 (5.3) |
| | TenGP | 0.1 (11.7) | 0.0 0.0 (8.8) | – | 0.0 0.0 (8.2) | 0.1 (9.8) | 0.0 0.0 (8.3) | – | 0.1 0.3 (8.2) |
| | PySR | 96.3 (3.0) | 41.7 44.1 (5.1) | – | 37.5 37.9 (3.5) | 87.7 (2.8) | 27.1 28.9 (4.3) | – | 29.2 29.1 (7.8) |
| NN | DiffLogic | 18.2 (110.4) | 1.1 2.2 (206.4) | 2.3 2.4 (1821.4) | 7.4 7.5 (256.2) | 15.7 (110.4) | 0.7 1.9 (206.4) | 2.6 2.6 (1821.4) | 6.3 6.3 (614.4) |
| | AI4EDA TNet | 26.0 (186.4) | 2.0 2.8 (192.4) | – | 8.7 8.7 (141.2) | 28.3 (186.4) | 0.3 1.6 (192.4) | – | 6.1 6.2 (499.6) |
| | NeuraLUT | 63.7 (6.4) | 79.0 70.5 (102.4) | – | 28.0 27.1 (51.2) | 64.3 (6.4) | 82.1 71.7 (102.4) | – | 27.2 26.4 (409.6) |
| | Boolformer | 100.0 (5.0) | 88.2 89.2 (18.1) | – | 19.1 19.0 (2.9) | N/A | N/A | N/A | N/A |
| | SATNet | 100.0 (40.0) | 78.1 72.7 (80.0) | 72.7 73.9 (160.0) | 41.3 40.2 (102.4) | 99.8 (40.0) | 71.5 66.9 (80.0) | 73.2 74.4 (160.0) | 32.7 31.8 (121.6) |
| Logic | ABC | N/A | N/A | N/A | N/A | 100.0 (5.8) | 100.0 0.8 (501.2) | – | 43.3 7.4 (717.6) |
| | BDD | 100.0 (56.8) | 100.0 0.5 (5718.9) | – | 21.5 10.9 (2416.9) | N/A | N/A | N/A | N/A |
| | Espresso | 100.0 (6.2) | 100.0 84.6 (22.4) | 100.0 93.7 (66.3) | 48.4 48.4 (10.8) | N/A | N/A | N/A | N/A |
| | Quine-McCluskey | 100.0 (6.5) | 100.0 0.5 (912.7) | – | 21.5 10.8 (363.7) | N/A | N/A | N/A | N/A |
| LLM | GPT-o3 | 90.1 (6.9) | – | – | 34.1 34.0 (2.8) | 88.0 (9.5) | – | – | 21.7 21.7 (89.9) |
| | GPT-4o | 1.6 (6.4) | – | – | 0.0 0.0 (5.9) | 2.1 (5.5) | – | – | 0.3 0.3 (23.3) |
| | GPT-o3 (PoT) | 97.1 (41.0) | – | – | 39.5 39.4 (30.2) | 95.0 (44.8) | – | – | 29.2 29.3 (102.6) |
| | GPT-4o (PoT) | 16.8 (21.7) | – | – | 5.1 5.1 (10.2) | 17.4 (21.5) | – | – | 5.4 5.3 (67.1) |

eling paradigms. Among traditional logic synthesis methods, surprisingly, Espresso maintains high accuracy for input scales ranging from 5 to 20. In contrast, ABC, Quine-McCluskey and BDD are suitable only for very small input sizes ($\leq 5$), with their accuracies sharply declining to nearly zero as the input scale increases to 10 (in5_out5 networks do not have a test split), indicating severe limitations in generalization. Most symbolic regression and neural network methods experience accuracy decline as scale increases, with Difflogic and GPlearn capable of handling larger datasets.

Based on the comparison of bit-wise accuracy and sample-wise accuracy shown in Figure 4(b), with an input size of 10, output size of 10, and under noise-free conditions, we observe the following: Bit-level accuracy tends to be higher due to the tolerance inherent in bit-wise local matching, whereas sample-wise accuracy decreases because of the stringent requirement for complete column-wise matching. This reveals a common issue among methods, wherein accumulated bit-wise errors frequently lead to entire column mismatches. Notably, Espresso, NeuraLUT and SATNet maintain good accuracy and generalization performance at both granularity levels.

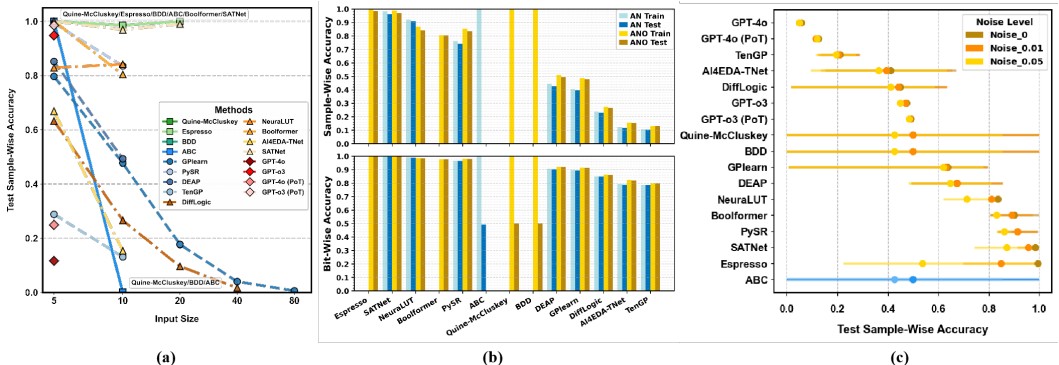

(a)  (b)  (c)

Figure 4: The correlation between the accuracy and different factors for each tested method. (a) Accuracy on logic networks with varying input sizes (without noise); (b) Sample-wise and bit-wise accuracy on AN and ANO datasets (input = 10); (c) Accuracy on logic networks with varying noise levels (average on input=5/10). In (a) and (c), the accuracy is measured on the synthetic ANO dataset for most methods, except for ABC, which is measured on synthetic AN dataset due to its limited applicability.

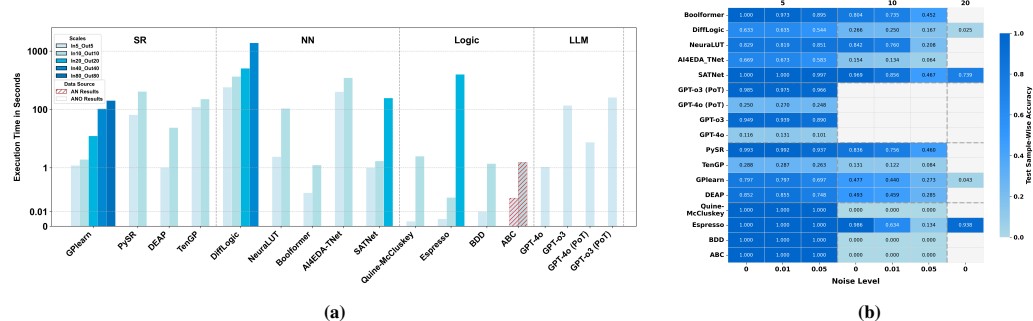

(a)  (b)

Figure 5: Comparison of baseline methods. (a) Computational efficiency under different scales; (b) Per-output correctness under varying input sizes and noise levels. In (a) and (b), The ABC* method only supports synthetic AN data, while the other 16 methods are evaluated on synthetic ANO.

In the noise scenario depicted in Figure 4(c), symbolic regression methods exhibit comparatively stronger robustness under noise partly because their search focuses on discovering compact structural patterns rather than exactly fitting every individual sample. As a result, small perturbations do not necessarily alter the preferred structural solutions. In contrast, logic synthesis methods are built to ensure exact consistency with the input truth table: every provided input–output assignment must be satisfied. When the truth table contains noise, these corrupted entries act as constraints, causing the synthesized expressions to deviate from the ground-truth function as the noise level increases.

### 4.4 ANALYSIS OF EFFICIENCY, AND FIDELITY

Figure 5(a) evaluates the time efficiency of the baseline methods on the synthetic dataset. From this figure, SR methods show a substantial increase in runtime (from $10^0$ to $10^3$ seconds), indicating high sensitivity to the input scale. This is because SR relies on a single-output iterative search mechanism, generating only one expression per run; consequently, as the input scale increases, the search space expands rapidly. Traditional logic synthesis methods, leveraging rule-based derivation mechanisms, simultaneously generate multiple outputs in a single run or parallel manner, thus exhibiting the shortest and most stable processing time at small scales (e.g., in5_out5).

Figure 5(b) reports sample-wise fidelity, defined as accuracy measured against the provided (potentially noisy) truth table, reflecting consistency with observed data rather than recovery of the original clean expressions. From this figure, we observe that neural network–based methods are more sensitive to noise, symbolic regression methods are more affected by increasing input scale, and logic synthesis methods degrade primarily due to their limited generalization capability.

## 5 RELATED WORK

The present research lies at the intersection of symbolic regression, logic synthesis, and AI benchmarking.

**Symbolic Regression.** Symbolic Regression (SR) involves identifying a mathematical expression in symbolic form that best fits a given dataset (Makke & Chawla, 2024) . The search space for SR encompasses all possible expressions, which is infinite and complex. Genetic Programming (GP) has been the predominant method for SR since its popularization by Koza (Koza, 1992) . In GP, a population of candidate expressions is evolved over generations using operations like selection, crossover, and mutation to enhance their fit to the data.

Nevertheless, the SR community has traditionally focused on finding functions in continuous domains, using a basis set of arithmetic operators $(+, -, \times, \div)$, analytic functions $(\sin, \cos, \exp, \log)$, and real-valued constants. Consequently, widely used SR benchmarks, such as Koza's original SR test suite (Koza, 1992) and the SRBench suite (Cava et al., 2021), primarily feature regression tasks over real numbers. The evaluation of SR for discrete or Boolean domains remains underexplored, and a standardized, large-scale benchmark for logical discovery has been a notable omission in the field. Our work directly addresses this gap.

**Logic Synthesis.** Logic Synthesis is a core component of Electronic Design Automation (EDA) that transforms an abstract specification of a digital circuit's behavior, typically a Register-Transfer Level (RTL) description, into an optimized logic gate implementation (Kurup & Abbasi, 1997; Reese & Thornton, 2006) . The primary goals of logic synthesis are to minimize metrics such as circuit area, power consumption, and signal delay, while strictly preserving the functional correctness specified by a Boolean function or network.

Well-known tools in this area, such as ABC (Brayton & Mishchenko, 2010) , are highly effective for optimization but operate under the assumption of a complete and exact logical specification. The benchmarks used in this field, like the EPFL combinational circuits (Amarú et al., 2015) and IWLS contest designs (IWLS, 2025), consist of established digital circuit designs. The evaluation paradigm is not concerned with learning from incomplete data samples or generalizing to unseen inputs, which are central objectives in machine learning. Our benchmark, in contrast, is specifically designed to evaluate these learning-based criteria, making it suitable for assessing modern approximate and data-driven methods that are not the focus of traditional LS.

**Neuro-Symbolic Methods and Benchmarking.** The integration of neural networks with symbolic reasoning, known as neuro-symbolic AI, is a rapidly growing field (Wan et al., 2024; DeLong et al., 2024) . These methods aim to combine the robust learning and pattern recognition capabilities of neural networks with the interpretability and reasoning power of symbolic systems. Many neuro-symbolic models are designed to address problems that involve discovering underlying rules or programs from data.

Progress in AI has often been driven by the introduction of high-quality, large-scale benchmarks (Deng et al., 2009; Wang et al., 2018). These benchmarks offer a shared basis for comparing various approaches, monitoring progress, and revealing the weaknesses of current methods. Our work advances this tradition by offering the first dedicated, large-scale benchmark for logical symbolic regression, which we believe will be crucial in developing and assessing the next generation of neuro-symbolic systems designed for logical discovery.

## 6 CONCLUSION

In this paper, we introduced LogicSR, a large-scale multi-granunlarity benchmark dedicated to logical symbolic regression. LogicSR addresses a critical gap between the fields of continuous symbolic regression and exact logic synthesis by creating a diverse suite of benchmark tasks. Using LogicSR, we conducted an extensive evaluation of 17 algorithms, spanning classical solvers, modern machine learning models, and large language models. Our analysis provides a clear map of the current state-of-the-art, revealing that existing methods face significant challenges with scalability, noise robustness, and logical operator compatibility. We hope to offer a robust foundation to benchmark progress, foster innovation across disparate research communities, and steer the future development of powerful and interpretable neuro-symbolic systems.

# 7 REPRODUCTIVITY STATEMENT

To ensure reproducibility, we provide the main results, algorithm pseudocode, methodological discussions, dataset quality analysis, and evaluation metric details in the appendix. The supplementary materials contain all additional scripts for data generation and experiments, the full dataset, complete result files for all runs, and documentation of implementation configurations. Upon publication, we will release the full codebase and datasets under an open-source license.

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

# A APPENDIX

## A.1 LLM USAGE STATEMENTS

We used LLMs only for language-level assistance, such as text polishing and generating preliminary pseudocode skeletons. All scientific ideas, algorithmic contributions, implementations, experiments, and the final manuscript were conceived, written, and verified by the human authors. LLMs were not involved in forming the paper's technical content or structure.

## A.2 GENERATING A SINGLE SMALL-SCALE LOGIC NETWORK

We formally describe the ground-truth small-scale logic network generation process in Algorithm 1. Before network construction, the algorithm specifies the network specifications and initializes the input nodes. It then iteratively generates $M_j$ gates, and finally ensures $O_j$ outputs through a designated output selection process.

Each iteration consists of two stages (lines 5–29): *generating and connecting a new node* and *computing truth vectors*.

**Generating and Connecting a New Node**. In the connection node generation stage, a new node $v_i$ with gate type $g_i$ is created (line 6). The candidate pool $\mathcal{C}$ is set to the dangling nodes if dangling number suffices for the remaining outputs; otherwise, all nodes can be chosen as candidate (line 7). Random inversion flags $(inv_0, inv_1)$ for $v_i$'s parenting nodes $(c_0, c_1)$ are initialized (line 8). Then, the first parent $c_0$ of $v_i$ is sampled from the pool, weighted by priority, and removed from the (copy of the) pool. Then a weight sequence for sampling within $\mathcal{C}$ is also initialized (line 9).

After that, the algorithm enters the stage for selection of the second parent $c_1$ (lines 10-20), and weights are adjusted for every candidate node $v_X \in \mathcal{C}$. There are three cases: 1) If $v_x$ is not an ancestor of $c_0$, its weight is directly set as the priority (lines 11-12); 2) If it appears only in one polarity on $c_0$'s ancestor path, alignment of $v_x$ with its previous polarity is enforced, and weight is set by a decayed factor priority$(v_x)/p_0$ (lines 13-15); 3) Otherwise, the priority is set as zero for the node, which must be trivial under this case (lines 16-18).

**Computing Truth Vectors**. After sampling both of the parents, the truth vector of $v_i$ is computed by $(\text{truth}(c_0)^{inv_0}, \text{truth}(c_1)^{inv_1})$. If the truth table is trivial, resampling is repeated until a nontrivial configuration is found or the maximum number of trials is reached (lines 21-25). Once condition met, $v_i$ is connected to $c_0, c_1$, and ancestor set $\text{Anc}(v_i)$ of $v_i$ is set as $\text{Anc}(c_0) \cup \text{Anc}(c_1) \cup \{v_i\}$ (line 26). Dangling set $\mathcal{H}_j$ is updated by $\mathcal{H}_j \leftarrow \mathcal{H}_j \cup \{v_i\} \setminus \{c_0, c_1\}$, and $\mathcal{S}_j \leftarrow \mathcal{S}_j \cup \{v_i\}$. $v_i$'s layer is set to be greater than that of its parent nodes:

$$\text{layer}(v_i) = \max(\text{layer}(c_0), \text{layer}(c_1)) + 1. \tag{4}$$

Then, priority of $v_i$ is set by a layer-aware rule:

$$\text{priority}(v_i) = p_0 + \sqrt{\frac{\text{layer}(v_i)}{p_0}}, \tag{5}$$

while the priorities of its parents $c_0$ and $c_1$ are decayed by $p_0$ to prevent logical reuse (line 27). This scheme promotes both structural depth and diversity in node selection. Finally, the new node is added to the network, and the maximum layer depth of the network is updated (line 28).

After $M_j$ iterations, the structure of the network is complete. The output allocation step is then performed by Algorithm 2, which selects an exact number of $L_j$ outputs (line 30).

**Network Output Allocation**. The goal of *Network Output Allocation* is to determine, within an already constructed logic network, how to select and assign nodes as final outputs in a way that ensures both rationality and diversity. The procedure (Algorithm 2) begins by clearing existing outputs if the flag `clear_outs` is set (lines 1–2). Nodes are then classified into nontrivial nodes $\mathcal{N}$, trivial nodes $\mathcal{T}$, and the subset of nontrivial nodes which are dangling $\mathcal{D}$ (i.e., not used as inputs to any other gates). The number of non-dangling nontrivial nodes further required is computed as (line 2):

$$r = L_{req} - |\mathcal{D}|. \tag{6}$$

If the number of dangling nontrivial nodes is already larger than the required outputs, $L_{req}$ nodes are directly sampled from $\mathcal{D}$ by a weight that combines node priority and layer depth (lines 3-4):

$$\text{output\_weight} = \max\left(\text{priority}(v) + \min\left(\frac{\text{layer}(v) - depth/\beta_1}{\beta_2}, 0\right), \epsilon\right) \quad (7)$$

where $depth$ is the maximum layer depth in this network, and $\beta_1, \beta_2 > 1$ are decaying factors ($\beta_1 = 6, \beta_2 = 2$ for our datasets generation scales), $\epsilon$ a minor constant ($10^{-6}$ in our experiments).

Otherwise, if the dangling nontrivial nodes are insufficient (lines 5-8), then all nodes in $\mathcal{D}$ are assigned as output (line 6). The remaining are then filled first from non-dangling nontrivial nodes (line 7). If these nontrivial nodes are still insufficient, trivial nodes are used as the last choice, with preference given to dangling trivial before non-dangling trivial nodes (line 8). This procedure ensures exactly $L_j$ outputs are allocated following the preference order $\langle$ nontrivial-dangling, nontrivial-non-dangling, trivial-dangling, trivial-non-dangling $\rangle$, prioritizing structurally meaningful and non-redundant outputs while maintaining robustness in case of insufficient candidates.

Finally, each selected node is marked as an output, with a random inversion $inv$ applied to increase the diversity of logical forms (lines 10-12).

---

**Algorithm 1** Small-Scale Logic Network Generation.

---

**Require:** number of inputs $K_j$, outputs $L_j$, steps $M_j$, maximum loop repeats MAX_R
**Ensure:** constructed network $\mathcal{G}_j$
1: Initialize $K_j$ input nodes $\mathcal{I}_|$ with priority $p_0$; set network depth $depth \leftarrow 1$
2: Initialize set of all nodes $\mathcal{S}_j \leftarrow K_j$, dangling nodes set $\mathcal{H}_j \leftarrow \mathcal{I}_j$
3: Assign each new input in $\mathcal{I}_j$ a full truth vector of length $2^{K_j}$, covering all input combinations
4: Sample gate operations sequence $\{g_1, \ldots, g_{M_j}\}$ uniformly
5: **for** $i = 1$ to $M_j$ **do**
6:     Create node $v_i$ with gate type $g_i \in \{\text{AND}, \text{OR}\}$, add $v_i$ to $\mathcal{G}_j$
7:     Set candidate pool $\mathcal{C} \leftarrow \mathcal{H}_j$ if $|\mathcal{H}_j| \geq L_j - 1$, else $\mathcal{S}_j$
8:     Initialize inversion flags $(inv_0, inv_1) \sim \{0, 1\}^2$ at random
9:     Sample $c_0 \in \mathcal{C}$ by priority, update $\mathcal{C} \leftarrow \mathcal{C} \setminus \{c_0\}$; initialize candidate weights $w_\mathcal{C}$
10:     **for all** $v_x \in \mathcal{C}$ **do**                ▷ Assigning weights to candidates.
11:         **if** $v_x \notin \text{Anc}(c_0)$ **then**
12:             $w_\mathcal{C}(v_x) \leftarrow \text{priority}(v_x)$
13:         **else if** $w_\mathcal{C}(v_x)$ appears as only non-inverted (resp. inverted) ancestor on path $\text{Anc}(c_0)$ **then**
14:             $inv_{v_x} \leftarrow$ True
15:             $w_\mathcal{C}(v_x) \leftarrow \text{priority}(v_i)/p_0$
16:         **else**
17:             $w_\mathcal{C}(v_x) \leftarrow 0$
18:         **end if**
19:     **end for**
20:     Sample $c_1 \in \mathcal{C}$ by $w_\mathcal{C}$; set attempts $\leftarrow 0$     ▷ Sample the second parent node based on candidate weights, and initialize the parameter for the triviality check.
21:     **while** trivial(truth($v_i$)) **and** attempts $<$ MAX_R **do**     ▷ Trivial check.
22:         Resample $(c_0, c_1)$ from $\mathcal{C}$ by priority
23:         Compute truth($v_i$) with $(inv_0, inv_1)$ reinitialized
24:         attempts $\leftarrow$ attempts+1
25:     **end while**
26:     Connect $v_i$ to $(c_0^{inv_0}, c_1^{inv_1})$; update $\text{Anc}(v_i)$; update $\mathcal{H}_j, \mathcal{S}_j$
27:     Update layer($v_i$), priority($v_i$); decay priority($c_0$), priority($c_1$)
28:     $depth \leftarrow \max(\text{layer}(v_i), depth)$     ▷ Update network depth.
29: **end for**
30: Allocate outputs by Algorithm 2

---

## A.3 MERGING PRE-GENERATED LOGIC NETWORKS

Algorithm 3 outlines the process of large-scale network generation. The input of Algorithm 3 is a sequence of small scale sub-networks $\mathcal{A}$ designed manually, as long as the sum of their sizes

---

**Algorithm 2** Network Output Allocation.

---

**Require:** network $\mathcal{G}$, number of outputs required $L_{req}$
**Ensure:** network $\mathcal{G}$ with exactly $L_{req}$ outputs allocated
1: Identify set of nontrivial nodes $\mathcal{N}$; dangling nontrivial nodes $\mathcal{D} \subseteq \mathcal{N}$; trivial nodes $\mathcal{T}$
2: $r \leftarrow L_{req} - |\mathcal{D}|$      ▷ Compute how many additional outputs are required after exhausting all dangling nontrivial nodes.
3: **if** $r < 0$ **then**            ▷ The case where the number of dangling nontrivial nodes is sufficient.
4:     Select $L_{req}$ nodes from $\mathcal{D}$ by weighted sampling
5: **else**                                          ▷ Fill output nodes in order of priority.
6:     Assign all nodes in $\mathcal{D}$ as outputs
7:     Select $\min(r, |\mathcal{N} \setminus \mathcal{D}|)$ nodes from $\mathcal{N} \setminus \mathcal{D}$ by weighted sampling
8:     Select $\max(r - |\mathcal{N} \setminus \mathcal{D}|, 0)$ nodes from $\mathcal{T}$, prioritizing dangling ones
9: **end if**
10: **for** each selected node $v$ **do**
11:     Mark $v$ as output with random inversion $inv \sim \{0, 1\}$
12: **end for**

---

$(K_j, L_j, M_j)$ does not exceed that of the large network's expected size respectively. The decayed priorities during the small-scale network generation are preserved, in order to give newly generated gate nodes a higher chance of being connected. Inputs $(K, L, M)$ represent the required number of inputs, outputs and internal gates, respectively, and $T < 2^K$ is the length of the partially preserved truth vector for each node.

During initialization, Algorithm 3 first computes and adds the required new input nodes $K_{\text{new}}$ (with base priority $p_0$ and partial truth vectors of length $T$), then initializes two sub-network usage counters $D_s$ and $D_c$, and finally sets the node sets $S$ and $H$ to the input nodes.

The main loop (lines 6-20) iteratively generates $M$ new gates. In each iteration, the key steps include selecting the parents of the new node, integrating the new node into the network, and updating the corresponding statistics.

**Parent Node Selection.**   For each newly generated gate node $v_i$ with gate type $g_i$ (line 7), the candidate pool is initialized by dangling node set $\mathcal{H}$ if number of dangling nodes exceeds the number of expected outputs $L$, or $S$ otherwise (line 8).

The algorithm then enters the parenting node selection stage (lines 12-16). Algorithm 4 is used for this parent node sampling process.

In Algorithm 4, $c_0$ is sampled from pool by a weight following power-law distribution, to combine selecting less-used nodes and structurally complex ones:

$$w(v) = \text{trivial}(v) \cdot \Big( \text{priority}(v) + \sqrt{|\text{Anc}(v)|/p_0} \Big)^{\alpha} \tag{8}$$

where $|\text{Anc}(v)|$ denotes the size of node $v$'s ancestor set, and $\alpha > 1$ is an amplification factor. Trivial nodes are specially set with a zero weight to prevent them from being selected (i.e. $w(v) = 0$).

After returning parent nodes by Algorithm 4, truth vector of $v_i$ is computed by truth of its parents (line 22). If $\text{truth}(v_i)$ is trivial, resampling continues until either a valid nontrivial truth table is obtained or the retry limit is exceeded (lines 3-6). In the latter case, $v_i$ is marked as trivial and will not contribute to any other nodes' formation. The subroutine finally returns $v_i$ with its truth vector computed as well as its parents and related sub-networks for post-processing (line 25). Getting node $v_i$, Algorithm 3 loops until $\text{truth}(v_i)$ is nontrivial or it reaches a maximum trial time.

**Integration of the New Node and Statistics Update.**   The post-processing stage then connects $v_i$ to the network, decays parent node priorities by $p_0$, updates ancestor sets of $v_i$, and sets the priority for the new node using the layer-aware strategy in Equation 5(lines 17-18). After that, dangling and entire node set are also updated the same way described in small-scale logic network generation, and essential counters of sub-networks are updated (line 19).

If the second subnetwork $\mathcal{G}_k$ returned by Algorithm 4 is not null, which means both the parents are selected from subnetworks, then $D_c$ will not be increased for both $\mathcal{G}_j$ and $\mathcal{G}_k$. If only $\mathcal{G}_j$ is not null, we update $D_c(\mathcal{G}_j) \leftarrow D_c(\mathcal{G}_j) + 1$ as it is connected to the main structure. Finally, $v_i$ is added to $\mathcal{G}$.

After $M$ iterations, the merged structure is complete. Outputs are allocated following Algorithm 2 (line 21).

---

**Algorithm 3** Large-Scale Logic Network Construction.

---

**Require:** $\mathcal{A} = \{\mathcal{G}_j(K_j, L_j, M_j)\}$; target scale $K, L, M$; truth vector reserved length $T < 2^L$; maximum repeat times $R_{\text{inner}}, R_{\text{outer}}$
**Ensure:** Merged network $\mathcal{G}(K, L, M)$
1: $K_{\text{new}} \leftarrow K - \sum K_j$
2: Initialize $K_{\text{new}}$ inputs $I$, add to $\mathcal{G}$ with priority $p_0$
3: Give each new input a partial truth vector of length $T$, covering a subset of input combinations
4: Initialize net select count $D_s \leftarrow 0$, connect count $D_c \leftarrow 0$
5: Initialize dangling set $\mathcal{H} \leftarrow \mathcal{I}$, node set $\mathcal{S} \leftarrow \mathcal{I}$
6: **for** $i = 1$ to $M$ **do**                                      ▷ Node generation and connection.
7:     Create $v_i$ with gate type $g_i \sim$ Connectives uniformly sampled
8:     $\mathcal{P} \leftarrow \mathcal{H}$ if $|\mathcal{H}| \geq L - 1$, else $\mathcal{S}$
9:     $p_{\text{pool}} \leftarrow \frac{|D_s|}{|\mathcal{A}|}, p_{\text{net}} \leftarrow \max\left(\epsilon, \sqrt{\frac{|\mathcal{I}|}{K_{\text{new}}}}\right)$
10:     $u_n \leftarrow (|\mathcal{A}| - |D_c|) \geq (M - i - 1)$
11:     attempts $\leftarrow 0$, initialize $\text{truth}(v_i) \leftarrow \{0\}^T$
12:     **while** trivial($\text{truth}(v_i)$) and attempts $< R_{\text{outer}}$ **do**       ▷ Trivial check.
13:         Initialize random inversions $(inv_0, inv_1) \sim \{0, 1\}^2$
14:         Derive $(v_i, c_0, c_1, \mathcal{G}_j, \mathcal{G}_k)$ by Algorithm 4
15:         attempts $\leftarrow$ attempts $+ 1$
16:     **end while**
17:     Update $\text{Anc}(v_i), \text{layer}(v_i), \text{priority}(v_i)$
18:     Decay priorities of $c_0, c_1$ by $p_0$
19:     Update $D_s, D_c, \mathcal{H}, \mathcal{S}$; add $v_i$ to $\mathcal{G}$
20: **end for**
21: Allocate outputs by Algorithm 2

---

### A.4 Quality Evaluation of Synthetic Datasets

In this section, we first provide a qualitative analysis of data quality by examining how the synthetic dataset reflects real-world logic, and further assess the diversity of the synthetic data through the uniqueness of truth tables across subnetworks.

### A.4.1 Structural Fidelity and Diversity of Synthetic Logic Networks

In real-world applications, most logic exists in the form of networks rather than isolated formulas. Moreover, many mainstream logic synthesis methods (such as ABC) can only process inputs in the form of logic networks (AIGs). Therefore, we generate our logic datasets based on network structures, leveraging network connectivity to better control the level of complexity.

In principle, there is no strict limitation on the ratio between the number of gates and the number of input variables. For example, in the EPFL benchmark suite (Amarú et al., 2015), circuits may contain only a few inputs but tens of thousands of gates. Nevertheless, to ensure that our test data remain closer to practical circuit designs and maintain a reasonable scale, we follow the structural characteristics of the ISCAS-85 benchmark circuits (Brglez et al., 1989). ISCAS-85 circuits have been widely used in logic synthesis and testing research, featuring moderate sizes (ranging from tens to thousands of gates), balanced input and output distributions, and clear hierarchical structures. These characteristics make ISCAS-85 a more realistic and structured basis for generating test datasets.

---

**Algorithm 4** Parent Node Selection based on Trivial Check.

---

**Require:** $v_i, \mathcal{P}, \mathcal{A}, p_{\text{pool}}, p_{\text{net}}, u_n, D_s, D_c, inv_0, inv_1, R_{\text{inner}}$
**Ensure:** Parenting node pair $(c_0, c_1)$
 1: attempts $\leftarrow 0$
 2: **while** attempts $< R_{\text{inner}}$ **do**
 3:      **if** $\neg\text{trivial}(\text{truth}(v_i))$ **then**               ▷ The case where the truth table is already nontrivial.
 4:          $v_i \leftarrow (c_0{}^{inv_0}, c_1{}^{inv_1})$
 5:          **break**
 6:      **end if**
 7:      Set two possible sub-networks $\mathcal{G}_j, \mathcal{G}_k$ to null
 8:      from_pool $\leftarrow (\text{rand}() < p_{\text{pool}} \wedge \mathcal{P} \neq \emptyset) \vee u_n$
 9:      **if** from_pool **then**
10:          $c_0 \leftarrow \text{Sample}(\mathcal{P}, w(v))$              ▷ Sample from $\mathcal{P}$ by weights.
11:          **if** $\text{rand}() > p_{\text{net}} \wedge \neg u_n$ **then**
12:              $c_1 \leftarrow \text{Sample}(\mathcal{P}, \text{priority}(v))$
13:          **else**
14:              $\mathcal{G}_j \leftarrow \text{Sample}(\mathcal{A}, \max(\nVdash_{D_c(\mathcal{G}_x)=0}, 1/D_s(\mathcal{G}_x)))$
15:              $c_1 \leftarrow \text{Sample}(\mathcal{G}_j, w(v))$
16:          **end if**
17:      **else**
18:          $\mathcal{G}_j, \mathcal{G}_k \leftarrow \text{Sample}(\mathcal{A}, \max(\nVdash_{D_c(\mathcal{G}_x)=0}, 1/D_s(\mathcal{G}_x)), 2)$
19:          $c_0 \leftarrow \text{Sample}(\mathcal{G}_j, w(v))$
20:          $c_1 \leftarrow \text{Sample}(\mathcal{G}_k, w(v))$
21:      **end if**
22:      $\text{truth}(v_i) \leftarrow \text{truth}(c_0)^{inv_0} \odot_{g_i} \text{truth}(c_1)^{inv_1}$
23:      attempts $\leftarrow$ attempts $+ 1$
24: **end while**
25: **return** $v_i, c_0, c_1, \mathcal{G}_j, \mathcal{G}_k$

---

### A.4.2   EVALUATION OF DIVERSITY FOR SYNTHETIC DATASET

In this section, we assess the diversity of the generated logic networks by analyzing truth-table uniqueness as the primary performance metric.

Table 4 summarizes all synthetic datasets along with their corresponding generation parameters. In the Generation Basis column, each tuple (from left to right) specifies: the number of inputs of each sub-network, the number of outputs, the number of sub-networks used to assemble a larger network, and the number of gate nodes within each sub-network.

Trivial vectors are included in the denominator but excluded from valid unique patterns. The global uniqueness values reported in Table 4 are computed for each sub-network size, and the overall distributions are visualized in Figure 6, where most configurations exhibit uniqueness ratios above 0.8.

### A.4.3   DISCUSSION ON ALGORITHM DESIGN FOR NETWORK SYNTHESIS

In this section, we systematically analyze its performance under varied settings, demonstrating both of its flexibility and boundaries.

**Input–Output Scale.** For synthetic datasets, we adopt symmetric input–output scales (equal numbers of inputs and outputs). This choice ensures consistent complexity scaling across tasks, enables fair comparison among multi-output settings, and provides a controlled environment for evaluating generalization. This design is consistent with common practices in prior work (Venere et al., 2024; Jiang et al., 2020) and also complements industrial benchmarks, which typically lack symmetric circuit structures (Amarú et al., 2015; Brglez et al., 1989).

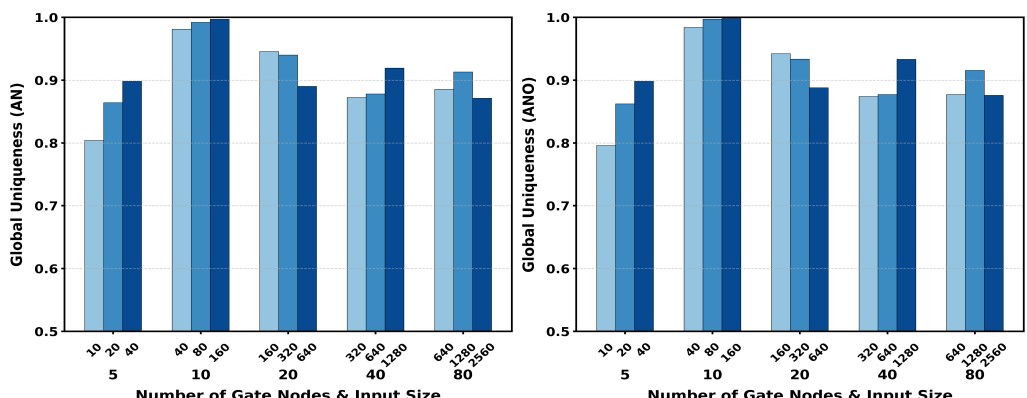

Figure 6: Global uniqueness across different AN/ANO configurations.

Table 4: Uniqueness ratio across input-output scales and network sizes in the LogicSR project.

| Scale | Number of Logic Gates | AN Uniqueness | ANO Uniqueness | Number of Networks | Generation Basis |
|---|---|---|---|---|---|
| in5_out5 | 10 | 0.8040 | 0.7960 | 100 | - |
| in5_out5 | 20 | 0.8640 | 0.8620 | 100 | - |
| in5_out5 | 40 | 0.8980 | 0.8980 | 100 | - |
| in10_out10 | 40 | 0.9810 | 0.9840 | 100 | - |
| in10_out10 | 80 | 0.9920 | 0.9970 | 100 | - |
| in10_out10 | 160 | 0.9970 | 0.9980 | 100 | - |
| in20_out20 | 160 | 0.9455 | 0.9420 | 100 | (5,5,3,30) |
| in20_out20 | 320 | 0.9400 | 0.9335 | 100 | (5,5,3,80) |
| in20_out20 | 640 | 0.8900 | 0.8880 | 100 | (8,8,2,170) |
| in40_out40 | 320 | 0.8725 | 0.8740 | 100 | (6,6,6,40) |
| in40_out40 | 640 | 0.8778 | 0.8770 | 100 | (6,6,6,90) |
| in40_out40 | 1280 | 0.9190 | 0.9330 | 50 | (5,5,7,150) |
| in80_out80 | 640 | 0.8855 | 0.8770 | 50 | (8,8,8,60) |
| in80_out80 | 1280 | 0.9130 | 0.9155 | 50 | (9,9,8,130) |
| in80_out80 | 2560 | 0.8710 | 0.8758 | 50 | (10,10,7,310) |

Our experiments further validate that symmetric input–output configurations produce more stable and predictable circuit structures. As shown in Figure 7, asymmetric settings (in $\neq$ out) result in lower truth-table uniqueness and substantially larger depth variance.

The decrease in global uniqueness under input-heavy configurations (inputs > outputs) is expected. With fewer output channels, many internally generated nodes produce distinct truth vectors that cannot be expressed at the output layer, limiting the maximum achievable uniqueness and naturally keeping it below 1. Conversely, when outputs exceed inputs, the consistently high uniqueness reflects the generator's inherent logical diversity and scalability.

**Hyperparameter Analysis of the Two-Stage Generator.** We conducted analysis on 3 key hyperparameters that influence the merging behavior and the overall network generation process. The amplification factor $\alpha$, derived from the weighting function in Equation 8, is evaluated on networks with 20 inputs, 20 outputs, and 640 logic gates. The priority decay coefficient $\mu$, which controls the decay rate of parent node priority during node expansion, is analyzed on networks with 10 inputs and 10 outputs. Finally, the effect of the remaining steps, defined as the difference between the target gate count $M$ and the total number of gate nodes already provided by the merged subnetworks $\sum_j M_j$, is studied on networks with 40 inputs, 40 outputs, and 640 logic gates.

From the experimental results, we observe the following: 1) $\mu$: A stronger decay produces deeper network structures but may reduce uniqueness, while the default decay setting achieves the best balance between depth and uniqueness. 2) $\alpha$: The default configuration ($\alpha = 1.5$) yields the highest uniqueness and the greatest depth, with the lowest FLV. This indicates that using an exponential weighting ($\alpha > 1$) is essential for emphasizing deeper nodes, thereby increasing structural com-

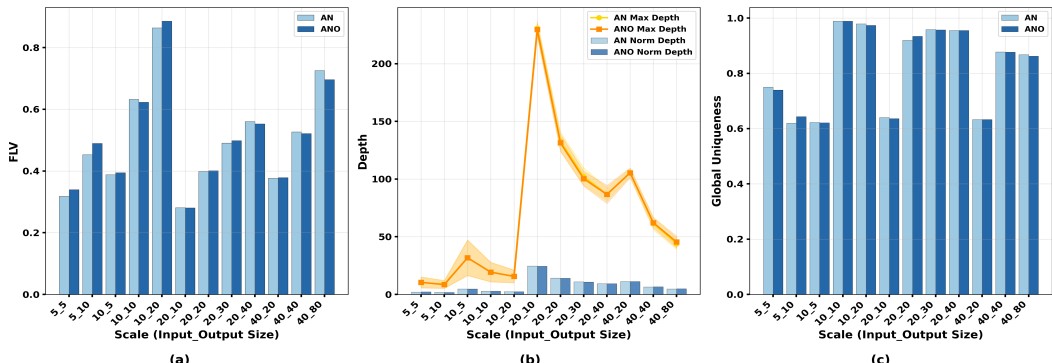

Figure 7: Comparative analysis of FLV (a), depth (b), and uniqueness metrics (c) for AN and ANO across asymmetric configurations.

Table 5: Comparison of structure metrics for different parameter settings.

| Hyperparameter | Settings (Param. Value) | FLV | Max_Depth | Normalized_Depth | Global_Uniqueness |
|---|---|---|---|---|---|
| | Linear (1) | 0.4885 | 122.54 | 13.08 | 0.901 |
| $\alpha$ | Default (1.5) | 0.4000 | 132.80 | 14.18 | 0.928 |
| | Strong Amplify (3) | 0.4838 | 123.82 | 13.22 | 0.912 |
| | No Decay (1) | 0.7482 | 17.97 | 2.660 | 0.986 |
| $\mu$ | Default (3) | 0.6214 | 19.51 | 2.894 | 0.990 |
| | Stronger Decay (9) | 0.6271 | 21.75 | 3.236 | 0.989 |
| | Large (480) | 0.4210 | 90.12 | 9.58 | 0.982 |
| Remaining Steps | Medium (300) | 0.5236 | 73.18 | 7.78 | 0.978 |
| | Small (100) | 0.6562 | 51.48 | 5.47 | 0.869 |

plexity while preserving diversity. 3) The Number of Remaining Steps: Configurations with a large number of remaining steps produce the lowest FLV, the maximum depth, and the highest uniqueness. This demonstrates that combining partial subnet reuse with a substantial amount of newly generated gates reduces structural repetition and maintains sufficient depth.

Other hyperparameters, such as the initial value of $p_0$, were also tested. However, within a reasonable range $[1, 12]$, all metrics remain largely unchanged, and therefore the corresponding results are omitted.

**Extension to a Broader Set of Operators.** Our algorithms are flexible and can be extended to larger operator sets by multiple ways shown in Table 6.

Table 6: Logic types and their extension schemes.

| **Logic Type** | **Extension Scheme** | **Example Operations** |
|---|---|---|
| Multi-valued (ternary) | Co-domain: $\{0,1\}^m \rightarrow \{0,1,2\}^m$ | Post algebra: MIN, MAX, Cycle |
| Fuzzy logic | Co-domain: $\{0,1\}^m \rightarrow [0,1]^m$ | t-norms/s-norms (Gödel, product) |
| Probabilistic Boolean | Mapping: $\{0,1\}^m \rightarrow \Delta(\{0,1\}^m)$ | Belief functions, Noisy-AND/OR |

## A.5 Considerations for Baseline-Related Experiments

### A.5.1 Additional Experimental Settings

In the experiments, for small-scale logic networks with 5 or 10 inputs, the truth vectors of the output variables were perturbed under three noise levels: 0.0 (noise-free), 0.01, and 0.05. For each input size, noisy datasets were generated and stored in separate folders, and these datasets were used directly for training and evaluation in the experiments. Our experimental protocol applied a per-instance timeout of 1200 seconds, and any instance that exceeded the time limit was excluded from the reported results. The selection of ANO and AN instances follows the criteria summarized in Table 7, while the full IWLS and BBM benchmark suites were used without modification.

Table 7: Experimental data count statistics for ANO and AN.

| Scale | Noise Level | Number of Networks |
|---|---|---|
| in5_out5 | 0 | 300 |
| | 0.01 | 300 |
| | 0.05 | 300 |
| in10_out10 | 0 | 150 |
| | 0.01 | 150 |
| | 0.05 | 150 |
| in20_out20 | 0 | 300 |
| in40_out40 | 0 | 150 |
| in80_out80 | 0 | 60 |

### A.5.2 CRITERIA FOR BASELINE SELECTION

In the experiments, different sets of baselines are selected for distinct datasets. Specifically, for datasets ANO and BBM, which include the complete gate set {AND, OR, NOT}, we used 16 baseline algorithms. Conversely, for datasets AN and IWLS, which contain only the gate set {AND, NOT}, 13 baseline algorithms were selected.

**Exclusion of Certain Algorithms on AN and IWLS Datasets.** We test only 13 baselines on the AN and IWLS datasets, excluding the algorithms *Quine-McCluskey*, *BDD*, *Espresso*, and *Boolformer*. The native minimal outputs of the *Quine-McCluskey* and *BDD* algorithms inherently include OR gates, being represented as Sum-of-Products and cube lists respectively. Similarly, *Espresso* and *Boolformer* algorithms explicitly include OR gates in their original expressions. Therefore, these four algorithms are more suitable for scenarios where the ground-truth is expressed using the {AND, OR, NOT} gate set. Forcing their outputs to map onto a pure {AND, NOT} gate set would require additional De Morgan transformations or logic synthesis, which could distort the original forms of the algorithm outputs.

**Rationale for ABC Evaluation Only on {AND, NOT} Gate Set.** The *ABC* algorithm is not tested on the ANO and BBM datasets, as it is only evaluated on datasets whose ground-truth expressions are represented using the {AND, NOT} gate set. Specifically, the internal structure of *ABC* being an And-Inverter-Graph (AIG), which consists solely of two-input AND gates, with inversion operations implemented through edge labeling. Thus, the {AND, NOT} gate set is sufficient to encompass all functionalities handled by *ABC* without impacting its generality. Moreover, most publicly available benchmarks and competitions related to *ABC* (e.g. HWMCC (Biere et al., 2024)) conventionally report results using the {AND, NOT} gate set. Maintaining consistency with this gate set allows our experimental results to directly align with these established benchmarks.

### A.5.3 DISCUSSIONS OF EXPERIMENTAL RESULTS

Performance variations across all methods on real-world and synthetic dataset, revealing the adaptability gap between structured benchmark tasks and complex logical interactions in practical scenarios. The synthetic data in LogicSR serve as controlled benchmarks, characterizing performance boundaries under clean logical conditions. In contrast, real-world datasets may contain more complex characteristics, including coupled multiple logical operators and implicit noise, imposing higher demands on the generalization and robustness of the evaluated methods.

### A.6 DETAILED DESCRIPTIONS OF BASELINES

**Traditional Logic Synthesis.** We select four representative methods from traditional logic synthesis: 1) *ABC* (Brayton & Mishchenko, 2010), a Logic Network-based synthesis and optimization tool; 2) *BDD* (Lee, 1959), a Binary Decision Diagram (BDD)-based simplification approach; 3) *Espresso* (Rudell & Sangiovanni-Vincentelli, 1987), an algebraic minimization tool implemented with the pyeda package; 4) *Quine-McCluskey* (Quine, 1952; for Symbolic Logic, 1947), a logic minimization method provided by the quine_mccluskey Python module.

**Symbolic Regression.** Within the symbolic regression category, we select four representative methods, all implemented based on Python libraries: 1) *PySR* (Cranmer, 2023), an open-source library for practical symbolic regression; 2) *DEAP* (Fortin et al., 2012), an evolutionary algorithm library

supporting tree-based genetic programming; 3) *GPlearn* (Stephens & Meeker, 2019), a scikit-learn-compatible genetic programming library featuring symbolic simplification; 4) *TenGP* (Arenas, 2022), a tensor-based Cartesian genetic programming method (Miller, 1999) that leverage efficient matrix representations.

**Neural Network-Based Methods.** The five neural network-based methods we select are as follows: 1) *DiffLogic* (Petersen et al., 2022), which performs gradient-based optimization on differentiable logical functions; 2) *AI4EDA_TNet* (W. et al., 2024), a neural network-based method for logic network transformation in electronic design automation (EDA) applications; 3) *NeuraLUT* (Cassidy et al., 2025), which uses supervised learning to train logic look-up tables (LUTs); 4) *Boolformer* (d'Ascoli et al., 2023), a transformer-based architecture specifically designed for Boolean logic reasoning. 5) *SatNet* (Wang et al., 2019), ,which integrates a differentiable SAT solver into neural networks to enable gradient-based logical reasoning.

**LLM-Based Methods.** Considering the significant potential demonstrated by large language models (LLMs) in reasoning tasks, we select two state-of-the-art LLMs: *GPT-4o* (Achiam et al., 2023), developed by OpenAI, and *GPT-o3* (OpenAI, 2024), an optimized, GPT-3-based high-performance model to perform prompt-based symbolic reasoning.

*1) LLM Few-Shot Prompt:* Based on few-shot learning, the LLM is prompted to act as a symbolic regression problem solver, aiming to automatically derive the simplest Boolean expression from a given truth table. The prompt comprises: (1) a role definition, (2) task description, (3) dynamically configured operators based on AN or ANO, (4) illustrative truth-table examples as few-shot teaching sequences, and (5) loaded complete truth table data ready for analysis.

---

**Few-Shot Prompt Example**

**System Prompt**
You are performing a logic symbolic regression task. Based on the truth table's input-output relationships, find the simplest boolean expressions.

**Task Description**
1. You are given a complete truth table with inputs and outputs.
2. For each output $y_K$, produce a correct and as-simplified-as-possible boolean expression.
3. Use variables $x_1$, ..., $x_{n_{inputs}}$ for inputs. Do not use variables outside this range.
4. Return exactly one line per output in the form `y_K = <expression>`.
5. Use explicit parentheses to indicate grouping.
6. If an output $y_K$ is a constant function, you may use the constant 0 or 1 for its expression.
7. No extra commentary besides the lines `y_K = <expression>`.
8. Optimization Goal (in order of priority):
   (a) Exact correctness on the provided truth table.
   (b) Minimal expression size (fewest gates / shortest simplified form).

---

**Operator Configuration**
1. Available Operators: {`allowed_operators`} (dynamically configured)
2. Usage Rules:
   (a) Use lowercase operator names only
   (b) Strictly use only the available operators listed above
3. Configuration Note: The operator set is dynamically controlled by the system configuration. Only the operators marked as available in the current configuration may be used in your expressions.

---

**Few-shot Examples: Teaching Sequences**
**Example 1 (2-input AND):**

| x1 | x2 | y1 |
|----|----|----|
| 0  | 0  | 0  |
| 0  | 1  | 0  |
| 1  | 0  | 0  |
| 1  | 1  | 1  |

Answer: y1 = x1 and x2
**Example 2 (2-input XOR):**

```
x1    x2    y1
 0     0     0
 0     1     1
 1     0     1
 1     1     0
```
Answer: y1 = (x1 and not x2) or (not x1 and x2)

---

**Data Preparation: Complete Truth Table**
1. Data Loading: The complete truth table data has been prepared and loaded for analysis.
2. Table Structure: $\{\texttt{n\_inputs}\}$ input variable(s) $(x_1..x_{\{\text{n\_inputs}\}})$ and $\{\texttt{m\_outputs}\}$ output variable(s) $(y_1..y_{\{\text{m\_outputs}\}})$.

Return results now

*2) LLM Multi-Step Program-of-Thought Reasoning Prompt:* The PoT prompt uses a two-step procedure. First, the model performs a chain-of-thought analysis of the truth table, identifying minterms and outlining simplification logic without producing final expressions. Second, the model converts this reasoning into executable Python code that derives and verifies the Boolean expressions using only the allowed operators. All reasoning appears as code comments, and only executable PoT code is returned.

**Multi-Step PoT Reasoning Prompt Example**

**Step 1: Chain-of-Thought Reasoning (Natural-Language Analysis)**

**System Prompt:** You are performing a logic symbolic regression task. Based on the truth table's input-output relationships, analyze and find the simplest boolean expressions.

**Task Description:**
1. You are given a complete truth table with $\{\text{n\_inputs}\}$ inputs $(x_1, x_2, ..., x_{\{\text{n\_inputs}\}})$ and $\{\text{n\_outputs}\}$ outputs $(y_1, y_2, ..., y_{\{\text{n\_outputs}\}})$.
2. Each row represents one input combination and its corresponding output values.
3. Your goal is to find the simplest boolean expression for each output $y_K$ that matches the truth table exactly.

**Chain-of-Thought Analysis Requirements:**
1. Identifies minterms / maxterms for each output $y_K$ (which input combinations produce output 1).
2. Explains grouping / simplification ideas (how to combine minterms into simpler expressions).
3. Mentions any sanity checks you would perform (e.g., verifying all rows match).

**Important:** Do NOT output final expressions or code yet. Just provide your reasoning. End with a short "Reasoning Summary" describing the best candidate expression for each output.

**Truth Table:** $\{\text{table\_md}\}$ |

**Chain-of-thought:**

---

**Step 2: Program-of-Thought Code Generation (Executable Python)**

**System Prompt:** You MUST perform Program-of-Thought reasoning and output ONLY executable Python code.

**Task:** Implement a `solve()` function that finds boolean expressions for $\{\text{n\_outputs}\}$ outputs based on the truth table.

**Data Format (provided in runtime context):**
- `X_data`: numpy array of shape $(\{\text{len(X)}\}, \{\text{n\_inputs}\})$ - each row is one input combination $[x_1, ..., x_{\{\text{n\_inputs}\}}]$
- `Y_data`: numpy array of shape $(\{\text{len(X)}\}, \{\text{n\_outputs}\})$ - each row is corresponding output values $[y_1, ..., y_{\{\text{n\_outputs}\}}]$
- Use the provided `X_data` and `Y_data` arrays (do NOT reconstruct them).

**Code Requirements:**
1. For each output $y_K$ (K = 1 to $\{\text{n\_outputs}\}$):
   (a) Identify minterms (rows where Y_data[:, K-1] == 1)
   (b) Derive the simplest boolean expression
   (c) Verify the expression matches ALL rows in the truth table
2. Use only allowed operators: $\{\texttt{allowed\_ops}\}$ (e.g., 'and', 'or', 'not').
3. Print exactly $\{\text{n\_outputs}\}$ lines in the format $y_K$ = <expression>.
4. All reasoning must appear inside Python comments + executable logic.

5. No natural-language text outside the code block.

**Chain-of-Thought Analysis Provided:**

{cot_text}

**Instruction:** Follow the Chain-of-Thought above and now produce the Program-of-Thought code.

**Few-shot Example (2-input AND):**

```
def solve():
    # Step 1: Identify minterms for y1 (rows where y1=1)
    minterms = []
    for i in range(len(X_data)):
        if Y_data[i, 0] == 1:
            minterms.append(tuple(X_data[i]))
    # Result: minterms = [(1, 1)]

    # Step 2: Derive expression from minterms
    # For AND: y1 = x1 and x2
    def f(x1, x2):
        return int(x1 and x2)

    # Step 3: Verify against all rows
    for i in range(len(X_data)):
        x1, x2 = X_data[i]
        assert f(x1, x2) == Y_data[i, 0]

    # Step 4: Print final expression
    print("y1 = (x1 and x2)")
```

**Output:** ONLY the final Python code implementing `solve()`, nothing else.

## A.7 ILLUSTRATION OF EVALUATION METRICS

### A.7.1 CALCULATION OF COMPLEXITY OF EXPRESSION

For methods that explicitly generate logical expressions, we measure expression complexity by averaging the number of Boolean operators AND, NOT, and OR in the output expressions when testing each baseline algorithm on a truth table.

The formula for calculating the number of operators is given as:

$$\frac{1}{N} \sum_{i=1}^{N} \text{ops}(E_i) \tag{9}$$

where $\text{ops}(E_i)$ represents the total operator count of the $i^{th}$ expression, defined as:

$$\text{ops}(E_i) = \text{NOT}_{num}(E_i) + \text{AND}_{num}(E_i) + \text{OR}_{num}(E_i) \tag{10}$$

In practical implementation, the *Quine-McCluskey* and *BDD* algorithms do not directly provide gate-level representations. Therefore, we calculated the total number of AND, NOT, and OR gates based on the *Quine-McCluskey* algorithm's Sum-of-Products form and the cube list generated by the *BDD* algorithm.

For the *Quine-McCluskey* algorithm, the gate count of the $i^{th}$ expression is given by:

$$\text{ops}(E_i) = \sum_{j=1}^{m_i} z_{i,j} + \sum_{j=1}^{m_i} \max(k_{i,j} - 1, 0) + \max(m_i - 1, 0) \tag{11}$$

where $m_i$ is the number of terms in the $i^{th}$ *Quine-McCluskey* output, $k_{i,j}$ denotes the total number of literals in the $j^{th}$ term of the $i^{th}$ expression, and $z_{i,j}$ is the number of negations required in the $j^{th}$ term of the $i^{th}$ expression.

For *BDD* algorithm, the gate count of the $i^{th}$ expression can be represented using the same formula (Equation 11), with differences only in the definitions of individual symbols: $m_i$ represents the

number of paths in the $i^{th}$ *BDD*, $k_{i,j}$ denotes the number of variables in the $j^{th}$ path of the $i^{th}$ expression, and $z_{i,j}$ represents the number of negations required in the $j^{th}$ path of the $i^{th}$ expression.

For neural network–based approaches, we consider five models in total, among which four (*DiffLogic*, *NeuraLUT*, *AI4EDA-TNet* and *SATNet*) only output predicted truth tables rather than explicit symbolic expressions. Consequently, their Boolean expression complexity cannot be derived directly. However, because their architectures bear a structural correspondence to certain Boolean operator compositions or LUT-like realizations, we can estimate their *equivalent Boolean circuit complexity* from the underlying architectural semantics. The details are as follows:

**1) DiffLogic.** *DiffLogic* stacks $L(k)$ layers of differentiable Boolean units, each containing $h(k)$ soft AND/OR operators. The first $L(k) - 1$ layers provide intermediate conjunctive/disjunctive compositions, while the final layer collapses to a LUT-like enumeration of all output configurations. Thus, the equivalent Boolean complexity is:

$$G_{\text{DiffLogic}}(k) = \big(L(k) - 1\big) \cdot h(k) + 2^k. \tag{12}$$

**2) NeuraLUT.** *NeuraLUT* parameterizes a lookup table with $2^k$ entries for each of the $k$ outputs. This corresponds to a canonical realization with $2^k$ minterms in the underlying Boolean function. Therefore, its equivalent Boolean complexity is:

$$G_{\text{NeuraLUT}}(k) = 2^k. \tag{13}$$

**3) AI4EDA-TNet.** *AI4EDA-TNet* augments a LUT-like output layer with $h$ hidden gated units, each corresponding to a logical operation or transformation. Its overall Boolean-equivalent complexity is estimated as:

$$G_{\text{AI4EDA-TNet}}(k) = h + 2^k. \tag{14}$$

**4) SATNet.** *SATNet* represents logical constraints using a set of learnable soft clauses. Its representational capacity is determined by the number of learnable clauses, which we take as its Boolean-equivalent complexity:

$$\chi \cdot (n_{\text{in}} + n_{\text{out}}) \tag{15}$$

Following the official *SATNet* repository [1], we set $\chi = 4$.

A.7.2 CALCULATION OF ACCURACY

To ensure a unified evaluation process across all methods, we first convert all outputs into a standard format of 0/1 truth table predictions. Specifically, for methods that inherently generate Boolean expressions, we evaluate these expressions on all input combinations to obtain a complete 0/1 truth table. For methods that directly output 0/1 sequences or predictions, we use their predictions directly.

Each method produces a predicted truth table $\hat{Y}$. We denote the number of input combinations (i.e., the number of samples in the truth table) as $S$, and the number of output variables as $M$. Under this notation, both the predicted and ground-truth truth tables are represented as:

$$\hat{Y}, Y \in \{0, 1\}^{S \times M}$$

where the $s$-th row corresponds to the $s$-th input combination, and the $m$-th column corresponds to the $m$-th output variable.

Based on the unified representation above, we define three evaluation metrics:

**1) Bit-Wise Accuracy:**

$$\text{bit\_accuracy} = \frac{1}{S \times M} \sum_{s=1}^{S} \sum_{m=1}^{M} \mathbb{I}(\hat{y}_{s,m} = y_{s,m}) \tag{16}$$

This metric calculates the proportion of correctly predicted bits over the entire truth table (total $M \times S$ bits).

---

[1] https://github.com/locuslab/SATNet

**2) Sample-Wise Accuracy:**

$$\text{sample\_accuracy} = \frac{1}{S} \sum_{s=1}^{S} \mathbb{I}(\hat{\mathbf{y}}_s = \mathbf{y}_s) \tag{17}$$

This metric counts a prediction as correct only if the entire output vector for a given input sample matches the ground-truth output exactly. It measures the proportion of input samples for which all output bits are predicted correctly.

**3) Per-Output Correctness (POC):**

$$\text{POC} = \frac{1}{M} \sum_{m=1}^{M} \prod_{s=1}^{S} \mathbb{I}(\hat{y}_{s,m} = y_{s,m}) \tag{18}$$

Following the single-output evaluation protocol commonly used in symbolic regression, and the exact-match criterion adopted in logic synthesis, we consider an output correct only if its predicted expression exactly matches the ground-truth function across all input combinations. Extending this definition to the multi-output setting, the POC metric reports the proportion of outputs whose expressions satisfy this exact-match requirement.

### A.7.3 SUPPLEMENTARY EXPLANATION OF TIME EFFICIENCY

Across all categories of methods, efficiency refers to the wall-clock time required to produce the final expressions (or predictions) for a given dataset instance. This definition is implementation-agnostic and applies uniformly to logic-synthesis, symbolic, neural, and LLM-based approaches.

**Logic-synthesis / symbolic-regression (non-learning) methods.** The reported cost corresponds to the single-pass solving time, i.e., loading the dataset, generating expressions from the training set, and producing predictions on the test set using the generated expressions.

**Machine-learning-based methods.** For *DiffLogic*, *AI4EDA-TNet*, *NeuraLUT* and *SATNet*, the reported cost is the end-to-end time, including both training and inference. This consists of training the neural network on the training set and then predicting on the test set using the trained model. For *Boolformer*, since a pretrained model is used, we report only the time to generate expressions from the pretrained model and the inference time on the test set.

**LLMs.** No fine-tuning is performed. The total time includes the API calls used to generate expressions from the training set.

### A.8 ANALYSIS OF EVALUATION APPROACHES FOR BASELINES

To comprehensively position our proposed benchmark, we examined the evaluation methodologies for baseline methods.

Among mature logic synthesis and symbolic regression methods, *ABC* (Brayton & Mishchenko, 2010) objectively demonstrates its advantages using clear performance metrics such as runtime, node count, and resource consumption. The *DEAP* framework (Fortin et al., 2012), on the other hand, evaluates its strengths in rapid prototyping and developing custom evolutionary algorithms by comparing code length, example richness, and clarity of practical implementation.

In neural network-based methods, *NeuraLUT* (Cassidy et al., 2025) emphasizes the evaluation of sparsity, efficiency, and adaptability to FPGA architectures. *Difflogic* (Petersen et al., 2022) utilizes multiple classic datasets, including the logical reasoning classification MONK series (MONK-1, MONK-2, MONK-3), the Adult Census dataset, the medical dataset of Breast Cancer, and the image datasets MNIST and CIFAR-10. Its evaluation metrics encompass classification accuracy, model parameter count, inference speed (CPU/GPU execution time), memory usage, and computational complexity (OPs/FLOPs). *ReduceLUT* (Cassidy et al., 2025) is evaluated on the Jet Substructure (JSC) dataset from particle physics classification tasks and the MNIST dataset, using metrics including physical lookup table (P-LUT) utilization, maximum circuit operating frequency ($F_{max}$), and neural network test accuracy. *Boolformer* (d'Ascoli et al., 2023) performs evaluations on synthetic

Boolean function datasets, real-world binary classification datasets from the PMLB collection, and Gene Regulatory Network (GRN) inference benchmarks. Its evaluation metrics include fitting accuracy and perfect recovery rate on synthetic datasets, F1 scores for real-world classification tasks, and dynamic accuracy, as well as structure-prediction-related binary classification metrics like F1 score and Youden's J in gene network tasks.

> **Key Differentiators:** Compared with these evaluation approaches, LogicSR maintains diverse data sources while introducing noise and constructing partially observable datasets, thus enabling a more effective assessment of algorithm robustness. Furthermore, our evaluation metrics are more refined, involving accuracy analysis at different granularities (bit-wise and sample-wise), along with the simplicity of expressions and time efficiency to assess the outputs. This multi-perspective approach provides a unified evaluation framework with broader applicability.

### A.9 DISCUSSION OF DOWNSTREAM APPLICATIONS

The task domains involved in LogicSR (such as gene regulation logic and circuit Boolean functions) can all be regarded as related application scenarios:

In biological Boolean networks (such as BBM), the regulatory relationships of genes are usually modeled using Boolean update rules to describe which regulatory factors activate or inhibit target genes under what combination of conditions. Real biological data are affected by sequencing errors, cellular heterogeneity, and experimental limitations, and thus can only observe sparse, noisy and discrete gene states. The significance of LogicSR lies in its ability to evaluate the restoration of logical dependencies under noise and partial observation. This ability can help propose more interpretable and analyzable regulatory hypotheses. Related works such as LogicNet (Malekpour et al., 2020) model gene expression as a probability distribution and jointly identify network topology and underlying logical structure; Research (Kadelka & Hari, 2025) systematically studied the accuracy of Threshold rules in Simple Boolean Threshold Networks relative to real biological logic.

In the EDA field, under the condition that a certain error is allowed, the Boolean functions recovered through learning can be used as interpretable approximations of circuit modules. For example, the average relative error of the FPGA approximate multiplier of SMApproxLib (Ullah et al., 2018) is within 0.126, and the complexity is approximately within 64 LUTs; Research (Iordanou et al., 2024) proposed a method for automatically generating a table data classification predictor circuit, with the number of logic gates controlled within 300 and an accuracy of 40-100% on 33 tasks.

In addition, EDA pipelines typically span multiple stages such as logic synthesis, optimization, and mapping. The ability to directly restore Boolean functionality from data means that some costly steps can be skipped, thereby enhancing design efficiency.

### A.10 EXTENDING LOGICSR BEYOND BOOLEAN FUNCTIONS AND BASIC OPERATORS

#### A.10.1 EXTENDING LOGICSR TO MULTIVALUED AND FUZZY LOGIC

LogicSR can be extended to non-Boolean functions by the following measures:

**1) Extended Operator Set:** Generalize binary operator sets to k-ary operators (e.g., ternary MIN/MAX) and continuous-value operators (e.g., t-norms for fuzzy logic). During the second-stage merging of the algorithm, parent node selection extends to sequential sampling without replacement of nodes, with other strategies preserved or adaptively optimized for specific settings.

**2) Generalized Data Representation:** Extend the binary value set $\{0, 1\}$ to discrete sets $\{0, \cdots, U-1\}$, expanding node representation vectors from $2^n$ to $U^n$ for multi-valued logic. For fuzzy or probabilistic logic, continuous sampling replaces enumeration with t-norm/t-conorm computation.

**3) Evaluation Metrics:** The current bit-wise and sample-wise accuracy metrics can be replaced by multi-class accuracy for multi-valued logic, or to continuous similarity measures for fuzzy logic.

### A.10.2 EXTENDING LOGICSR TO IMPLICATION OPERATORS

Scaling to more complex reasoning tasks with implications requires handling of relationships between logical rules, for example, $A \to B$ or chained rules $A \to B \to C$. Although these tasks fall under the category of rule-based reasoning, they can ultimately be encapsulated in a Boolean function: The current operator set is completely adequate in terms of expressive power, since $A \to B$ can be expanded using AND/NOT ($\neg A \vee B$). The following complementary methods could make LogicSR naturally extend to such tasks:

**1) Extending the Operator Set to Include Implication:** Implication ($\neg A \vee B$) could be directly added to the operator sets of LogicSR without any modifications, as it is a 2-ary operator, and every node could be regarded as an implication relation between its parenting nodes.

**2) Modeling with a Higher-Depth Network:** As multi-step implication could be represented by network structures with a higher depth, and LogicSR's two-stage synthesis framework could increase the network depth by including more gating nodes and input variables in the network structures, without introducing any new mechanisms.

### A.11 COMPARISON WITH OTHER BENCHMARKS

In this section, we systematically compared LogicSR with the current three major types of related benchmarks (Table 1) - the logic synthesis benchmark (EPFL, ISCAS, IWLS) (Amarú et al., 2015; Brglez et al., 1989; IWLS, 2025), and the continuous symbolic regression benchmark (LLM-SRBench, SRBench) The Benchmark (Shojaee et al., 2025; Cava et al., 2021) and the Boolean network benchmark (BBM, MCBF) in systems biology (Pastva et al., 2023; Subbaroyan et al., 2022) are summarized as follows:

The logic synthesis benchmark (EPFL (Amarú et al., 2015), ISCAS (Brglez et al., 1989), IWLS (IWLS, 2025)) only supports exact synthesis. The circuit structure is fixed and unadjustable, does not contain noise or some observable data, and cannot be used to evaluate generalization ability. Its evaluation metrics are limited to size/depth and do not involve expression recovery or learning performance.

Continuous symbolic regression benchmarks (LLM-SRBench (Shojaee et al., 2025), SRBench (Cava et al., 2021)) operate in the real number field with continuous operators ($+, -, \times, \sin, $ etc.), which are completely incompatible with Boolean logic semantics; Its accuracy index relies on continuous residuals (such as NMSE, $R^2$), and cannot be applied to the recovery task of scatter truth tables. Furthermore, the combinatorial explosion of Boolean logic expressions makes them significantly superior to continuous SR in structural scale and solution difficulty.

Although Boolean network benchmarks (BBM (Pastva et al., 2023), MCBF (Subbaroyan et al., 2022)) contain Boolean structures, their scales are fixed, and they lack adjustable operators, noise modeling, sampling mechanisms, and support for multi-task settings.

> **Key Differentiators:** Unlike the above three types of benchmarks, LogicSR offers for the first time an extensible input scale (5-80+), controllable network depth, structural complexity, and node sharing mode. Supports noise, partial truth table sampling, multi-output (MIMO) dependency structure, uniformly evaluates multiple types of methods, and cross-domain tasks. It exceeds the existing benchmarks in terms of scalability and complexity.

### A.12 LIMITATIONS AND FUTURE WORK

LogicSR still exhibits several limitations that provide opportunities for future investigation:

**1) Structural Constraints:** The binary fan-in design ensures validity but limits expressiveness for highly interconnected circuits, while the simplified operator set may not fully capture optimization challenges in industrial gate libraries (e.g., AOI, MUX). Extending to multi-fanin nodes and technology-mapped operators would better reflect industrial complexity.

**2) Scalability Constraints:** Evaluation remains constrained to practical-scale problems. Distributed frameworks and hierarchical generation could enable benchmarking of industrial ultra-large-scale systems (e.g., ASICs).

**3) Domain Applications:** While the benchmark focuses on general Boolean regression, the synthetic data generation currently lacks domain-specific inductive biases; in this work, we compensate for this by incorporating real-world datasets, but future extensions may further explore domain-knowledge-driven synthetic logic generation and richer forms of logic found in real systems, such as temporal logic in control and scheduling scenarios.

