# OpenReview forum: "LogicSR: A Unified Benchmark for Logical Discovery from Data"
_ICLR.cc/2026/Conference — ICLR 2026 Conference Desk Rejected Submission_

### Official Review · Reviewer_T9Hd · 2025-10-25

**Soundness:** 3
**Presentation:** 2
**Contribution:** 3
**Rating:** 6
**Confidence:** 2

**Summary:**

This paper introduces LogicSR, a unified benchmark for discovering logical expressions from data. The benchmark combines (i) curated real‑world problems from digital circuit design and biological Boolean networks (BBM), and (ii) a two‑stage synthetic data generator that first builds small logic networks and then merges them into larger ones. The authors evaluate 14 algorithms spanning traditional logic synthesis, symbolic regression, neural models, and LLMs, using metrics for in‑/out‑of‑distribution accuracy, expression complexity, and efficiency. Key findings are that classical logic synthesis excels on small scales but struggles to generalize or scale while neuro‑symbolic and SR methods generalize better but pay higher compute.

**Strengths:**

* The benchmark spans two practical domains (digital circuits from IWLS and biological Boolean networks from BBM) plus one synthetic domain, yielding a broad and well‑structured testbed across scales.
* The study evaluates methods comprehensively from rule‑based logic synthesis through symbolic regression and neural networks to state‑of‑the‑art LLMs, offering a rare cross‑paradigm view.

**Weaknesses:**

* While the paper motivates the task by its importance to EDA and biology, it remains unclear to non‑specialists what score on LogicSR signifies practical readiness in those domains. Mapping sample‑wise/bit‑wise accuracy to domain‑level utility would help readers judge how “promising for practice” current AI methods really are.

**Questions:**

* How do the IWLS (EDA) and BBM (biology) tasks compare to real‑world difficulty, and what do your scores imply about AI’s practical readiness?

---

> ### Author Response · Authors · 2025-11-22
>
> We thank the reviewer for the valuable feedback. We appreciate the opportunity to address these concerns and further improve the quality of our work. Below we provide analysis and clarifications in response.
> ________
> > **Q1: How do the IWLS (EDA) and BBM (biology) tasks compare to real‑world difficulty, and what do your scores imply about AI’s practical readiness?**
>
> **Response:** Thank you for this question.
>
> **IWLS and BBM Datasets.** The IWLS (EDA) dataset represents meaningful yet tractable Boolean circuits (5–16 inputs) that reflect real-world designs, including multi-output combinational logic, reconvergent fan-out patterns, and mixed control/data logic. The BBM dataset contains Boolean gene-regulatory networks derived from biological studies. Small networks (≤12 inputs) are fully enumerated to cover all discrete states, whereas larger ones are generated by simulated biological dynamics with both periodic and stochastic behaviors, ensuring realistic nonlinear transitions. All networks incorporate bit-flip noise injection to emulate biological variability.
>
> To illustrate what our scores imply about AI’s practical readiness, we present representative examples (ANO). We select cases with characteristic complexity (measured by number of gates) and corresponding sample-wise accuracy. “N/A’’ indicates that no valid output can be produced within the time limit.
>
> **1) Number of Gates:**
>
> |                    | in5_out5_noise 0.00 | in5_out5_noise 0.01 | in5_out5_noise 0.05 | in10_out10_noise 0.00 | in10_out10_noise 0.01 | in10_out10_noise 0.05 |
> |------------------|------------------|--------------------|--------------------|---------------------|-----------------------|-----------------------|
> | Boolformer    | 5.026            | 4.697              | 6.674              | 18.068              | 10.163                | 9.477                 |
> | GPT-o3 (PoT)  | 40.999           | 43.532             | 82.925             | N/A                 | N/A                   | N/A                   |
> | GPT-4o (PoT)  | 21.682           | 24.904             | 26.394             | N/A                 | N/A                   | N/A                   |
>
>
>
> **2) Sample-Wise Test Accuracy:**
>
> |      | in5_out5_noise 0.00 | in5_out5_noise 0.01 | in5_out5_noise 0.05 | in10_out10_noise 0.00 | in10_out10_noise 0.01 | in10_out10_noise 0.05 |
> |---------------|------------------|--------------------|--------------------|---------------------|-----------------------|-----------------------|
> | Boolformer    | 1.000            | 0.973              | 0.895              | 0.804               | 0.735                 | 0.452                 |
> | GPT-o3 (PoT)  | 0.985            | 0.975              | 0.966              | N/A                 | N/A                   | N/A                   |
> | GPT-4o (PoT)  | 0.250            | 0.270              | 0.248              | N/A                 | N/A                   | N/A                   |
> | NeuralLUT     | 0.829            | 0.819              | 0.851              | 0.842               | 0.760                 | 0.208                 |
> | SATNet        | 1.000            | 1.000              | 0.997              | 0.970               | 0.856                 | 0.467                 |
> | DiffLogic     | 0.633            | 0.635              | 0.544              | 0.266               | 0.250                 | 0.167                 |
> | AI4EDA_TNet   | 0.669            | 0.673              | 0.583              | 0.154               | 0.134                 | 0.064                 |
>
> **Interpretation.** Our results reflect the current upper bound of AI systems on higher-dimensional logical symbolic regression. Neural-network-based methods handle noise but struggle to scale. For LLM-based approaches, we observe clear degradation with increasing dimensionality. In the in5_out5 setting, GPT-o3 maintains high sample-wise accuracy under low noise, yet GPT-4o already exhibits limited functional reliability (<0.3 accuracy). At in10_out10, GPT-4o requires 140k+ tokens to generate the full set of expressions for a single truth table, and the resulting circuits exceed 6000 gates, leading to substantial complexity inflation and unstable inference. GPT-o3 similarly destabilizes when the reasoning chain becomes too long.
>
> **Summary.** While current models show some practicality on small-scale problems, they still encounter fundamental scalability bottlenecks. These limitations mirror the core challenges of real-world high-dimensional Boolean reasoning tasks, indicating that current AI systems are not yet fully ready for such scenarios.

---

> > ### Author Response · Authors · 2025-11-28
> >
> > >**W1: While the paper motivates the task by its importance to EDA and biology, it remains unclear to non‑specialists what score on LogicSR signifies practical readiness in those domains. Mapping sample‑wise/bit‑wise accuracy to domain‑level utility would help readers judge how “promising for practice” current AI methods really are. **
> >
> > **Response:**  Thanks for the comments. In EDA and biology fields, the definition of usability is different and dependent on its application scenarios, so there is no unified mapping relationship between accuracy and usability.
> >
> > In the EDA scenario, if the synthesized circuits target probabilistic or approximate behaviors, a relatively lower accuracy may be acceptable. In contrast, when the circuits are intended for precise control logic, the accuracy requirements become significantly higher. One application direction of error tolerance is approximate circuits [4,5]. The mean relative error of the 8-bit adder in EvoApprox8b [4] is within 16%, while that of the multiplier is within 12%, and the complexity is reported by power and delay. The average relative error of the FPGA approximate multiplier of SMApproxLib is within 0.126, and the complexity is approximately within 64 LUTs [5]; TinyClassifier [6] can automatically generate a table data classification predictor circuit, with the number of gates controlled within 300 and an accuracy of 40-100% on 33 tasks.
> >
> > In the biological scenario, one of the related tasks is GRN inference, whose goal is to predict whether a progenitor gene regulates a progenitor gene, and thus is applicable to metrics such as precision, recall, and AUPR [1, 7]. Reference [1] indicates that as long as AUPR or EPR is slightly higher than the random baseline, hypothesis generation can be supported. Researchs [3] studies whether simple approximations of Boolean models can approximate complex truths, pursuing higher bit-level consistency rates, and achieving mean agreement with biological rules (79.37%). In terms of complexity, Reference [2] points out that the controllability of real GRNs and the steady-state structure have no direct correspondence with the complexity of Boolean expressions.
> >
> > **References**
> >
> > [1] Pratapa, A., Jalihal, A. P., Law, J. N., Bharadwaj, A., & Murali, T. M. (2020). Benchmarking algorithms for gene regulatory network inference from single-cell transcriptomic data. Nature Methods, 17(2), 147–154.
> >
> > [2] Kadelka, S., et al. (2024). A meta-analysis of Boolean network models reveals design principles of gene regulatory networks. Science Advances, 10(2), eadj0822.
> >
> > [3] Kadelka, C., & Hari, K. (2025). Critical assessment of the ability of Boolean threshold models to describe gene regulatory network dynamics. PNAS Nexus, 4(8), pgaf228.
> >
> > [4] Mrazek, V., Hrbacek, R., Vasicek, Z., & Sekanina, L. (2017). EvoApprox8b: Library of approximate adders and multipliers for circuit design and benchmarking of approximation methods. Design, Automation & Test in Europe Conference (DATE).
> >
> > [5] Ullah, S., Murthy, S. S., & Kumar, A. (2018). SMApproxLib: Library of FPGA-based approximate multipliers. In Proceedings of the 55th Annual Design Automation Conference (DAC), 1–6.
> >
> > [6] Iordanou, K., Atkinson, T., Ozer, E., Kufel, J., Aligada, G., Biggs, J., Brown, G., & Luján, M. (2024). Low-cost and efficient prediction hardware for tabular data using tiny classifier circuits. Nature Electronics, 7(5), 405–413.
> >
> > [7] d’Ascoli, S., Renard, A., Papadopoulos, V., Bengio, S., Susskind, J. M., & Abbe, E. (2023). Boolformer: Symbolic regression of logic functions with transformers. 2nd AI for Math Workshop @ ICML 2025.

---

### Official Review · Reviewer_ZSc6 · 2025-10-29

**Soundness:** 3
**Presentation:** 2
**Contribution:** 3
**Rating:** 6
**Confidence:** 2

**Summary:**

Learning logical expressions from data is a critical task for interpretable AI and scientific discovery. However, existing research still lacks sufficiently comprehensive benchmarks for evaluating logical expression learning. In addition, current benchmarks are primarily designed under idealized conditions and do not account for noisy or incomplete data. In this work, the authors propose an automated method for generating benchmarks for logical expression learning across different scales and noise levels. They further evaluate 14 algorithms to assess their capabilities in inductively learning logical expressions.

**Strengths:**

The paper addresses a distinctive and important challenge in the machine learning and rule learning research communities. The proposed benchmarks enable rigorous evaluation of inductive reasoning capabilities.

**Weaknesses:**

The paper proposes a novel set of benchmarks. However, the organization of the manuscript makes it difficult to locate and assess the key information. For example, the complexity characteristics of the generated benchmarks are not clearly presented in the main content. In addition, more insightful comparisons with other similar benchmarks are missing. Please refer to the detailed questions below.

**Questions:**

1. How is the benchmark complexity systematically determined? What indicators are used to characterize and quantify the complexity? Moreover, is there any analysis of the network structure beyond simply reporting the number of input and output nodes?
2. In addition to accuracy, are there other metrics considered for evaluation, such as average recall or precision? If not, please provide justification.
3. How is noise introduced into the data, and what is the formal or mathematical definition of noise used in this work?
4. Are there any comparisons with other existing benchmarks in terms of scalability and complexity? If not, please elaborate on this aspect.

---

> ### Author Response · Authors · 2025-11-22
>
> We thank the reviewer for the valuable feedback. We appreciate the opportunity to address these concerns and further improve the quality of our work. Below we provide analysis and clarifications in response.
>
> _______________________________
>
> > **Q1: How is the benchmark complexity systematically determined? What indicators are used to characterize and quantify the complexity? Moreover, is there any analysis of the network structure beyond simply reporting the number of input and output nodes?**
>
> **Response:** Thank you for this valuable feedback. In the original version of LogicSR, we have already evaluated the algorithm's performance by the generated outputs' truth tables as stated in **Appendix A.4.2**: Average Uniqueness (per gate level then take average), Global Uniqueness (per input size level), and Max Repeat (dominant truth vector frequency). Following your suggestion, we added three structural feature evaluation metrics:
>
> **1) Fanout Layer Variance (FLV):** This metric is designed to measure the variance of fanout in each layer, and takes the mean across layers, indicating structural imbalance versus uniformity. A moderate FLV represents a structurally complex and balanced structure.
>
> **2) Maximum Depth:** Represents the longest path from inputs to outputs, reflecting circuit complexity.
>
> **3) Normalized Depth:** Scaled depth by circuit size (maximum_depth/log2(*Size*)), enabling cross-scale comparison.
>
> We adopted these three commonly used metrics following the best practices in related domains. The enhanced evaluation incorporating these structural metrics alongside the original truth-table analysis is presented below.
>
> | Config | In_Size | Out_Size | FLV    | Max_Depth | Normalized_Depth | Avg_Uniqueness | Global_Uniqueness | Max_Repeat |
> | ------ | ------- | -------- | ------ | --------- | ---------------- | -------------- | ----------------- | ---------- |
> | AN     | 5       | 5        | 0.3060  | 10.4233   | 2.1449           | 0.8553         | 0.7433            | 0.0047     |
> |        | 10      | 10       | 0.6226 | 19.5233   | 2.8973           | 0.9900           | 0.9883            | 0.0010      |
> |        | 20      | 20       | 0.4719 | 72.9933   | 8.3278           | 0.9252         | 0.9112            | 0.0030      |
> |        | 40      | 40       | 0.5063 | 72.1233   | 7.4624           | 0.8898         | 0.8592            | 0.0014     |
> |        | 80      | 80       | 0.5936 | 75.1733   | 7.0502           | 0.8898         | 0.8811            | 0.0021     |
> | ANO    | 5       | 5        | 0.3173 | 10.2933   | 2.1113           | 0.8520          | 0.7507            | 0.0053     |
> |        | 10      | 10       | 0.6298 | 19.3500     | 2.8705           | 0.9930          | 0.9900              | 0.0013     |
> |        | 20      | 20       | 0.4704 | 73.4233   | 8.3735           | 0.9212         | 0.9048            | 0.0028     |
> |        | 40      | 40       | 0.4989 | 72.1067   | 7.4691           | 0.8947         | 0.8568            | 0.0013     |
> |        | 80      | 80       | 0.5926 | 75.5600     | 7.0814           | 0.8894         | 0.8815            | 0.0000       |

---

> ### Author Response · Authors · 2025-11-22
>
> > **Q2: In addition to accuracy, are there other metrics considered for evaluation, such as average recall or precision? If not, please provide justification.**
>
> **Response:** Thank you for the question. In terms of accuracy evaluation, LogicSR employs two strict metrics: 1) Bit-wise accuracy (local correctness); 2) Sample-wise accuracy (global correctness of each output function).
>
> Recall is an metric in classification tasks, used to measure how many samples that are truly in the positive class have been successfully identified by the model. However, in Boolean symbolic regression, the output is a deterministic Boolean function, and each input combination has a unique label. Therefore, there is no positive/negative class semantics. Mathematically, Recall degenerates into a variation of accuracy. Precision is also related to positive and negative examples.
>
> Other benchmarks also adopt accuracy-related metrics. For example, in the symbolic regression (SR) literature, reference [1] evaluates performance using accuracy measures for continuous numerical SR (primarily $R^2$), while reference [2] employs continuous-error metrics such as MSE, MAE, and $R^2$.However, these metrics rely on real-valued outputs and continuous residuals, which do not align well with the task definition of Boolean function recovery, where evaluation is based on discrete, fine-grained truth-table matching.
>
> Furthermore, for the integrity of the project, we actually implemented the calculations of Precision, Recall and F1 Score in the code (see evaluate/metrics.py). Therefore, these metrics are optional, but we did not report them in the experimental results for the reasons mentioned above.
>
> **References**
>
> [1] Virgolin, M., & Krawiec, K. (2022). Contemporary symbolic regression methods and their relative performance. Proceedings of the Genetic and Evolutionary Computation Conference (GECCO), 235–244.
>
> [2] Kamienny, P.-A., Ieronymakis, A., Siems, J., La Cava, W., Jin, Y., & Schmidt, M. (2024). LLM-SRBench: Benchmarking large language models for symbolic regression. arXiv preprint arXiv:2402.18679.
>
> _______________________________
>
> >  **Q3: How is noise introduced into the data, and what is the formal or mathematical definition of noise used in this work?**
>
> **Response:** Noise is introduced via deterministic bit-flips on a fixed proportion of bits in each output's truth table.
>
> Formally, for each output function $y_j : \mathbb{B}^K \to \mathbb{B}$ ($\mathbb{B}$ denotes the Boolean domain), we randomly select exactly $\left\lfloor \eta \cdot 2^K \right\rfloor$ input combinations and flip the corresponding output value:
>
> $ỹ_j(x) = (x ∈ S_j)$ ?  $(y_j(x) ⊕ 1) : y_j(x)$
>
> where $\eta \in$ {$0, 0.01, 0.05$} is the noise rate, $S_j$ is a randomly chosen subset of the input space with $|S_j| = \left\lfloor \eta \cdot 2^K \right\rfloor$. This per-output corruption model captures systematic errors in experimental measurements and is commonly used in circuit testing and biological network inference.

---

> ### Author Response · Authors · 2025-11-28
>
> >**Q4: Are there any comparisons with other existing benchmarks in terms of scalability and complexity? If not, please elaborate on this aspect.**
>
> **Response:** Thank you for the comments. The comparison and summary with other benchmarks are as follows:
>
> | **Benchmark / Feature** | **Domain**            | **Scalability**                          | **#Samples**              | **Noise / Sampling** | **MIMO** | **Boolean** | **Metrics**                                   |
> |--------------------------|------------------------|--------------------------------------------|----------------------------|------------------------|---------|-------------|-----------------------------------------------|
> | EPFL [1]                 | Circuit                | Fixed                 | 23                         | ✗                      | ✗       | ✓           | Size / Depth                                  |
> | ISCAS’85/’89 [2]         | Circuit                | Fixed            | ’85: 10 + '89: 31          | Partial                | ✓       | ✓           | Size / Depth                                  |
> | IWLS’25 [3]              | Circuit                | Fixed                                      | 100                        | Partial                | ✗       | ✗           | Size / Depth                                  |
> | LLM-SRBench [4]          | Real-valued            | Scalable                                   | 239                        | ✓                      | ✗       | ✗           | NMSE, numeric precision, etc.                 |
> | SRBench [5]              | Real-valued            | Fixed                                      | 252                        | ✓                      | ✗       | ✓           | Complexity, R²                                |
> | BBM Boolean Models [6]   | Biological networks    | Fixed                                      | 245                        | Partial                | ✓       | ✓           | Function-level metrics                         |
> | MCBF [7]                 | Biological networks    | Fixed     | 2687                       | ✓                      | ✓       | ✓           | Boolean complexity, sensitivity, enrichment p-value |
> | **LogicSR (ours)**       | **Boolean networks**       | **Scalable (5–80+)**                       | **1000+**                  | **✓**                  | **✓**   | **✓**       | **Bit/Sample accuracy, complexity**            |
>
> 1) The **logic synthesis benchmarks** (EPFL, ISCAS, IWLS) [1-3] only support exact synthesis. The circuit structure is fixed and unadjustable, does not contain noise or some observable data, and cannot be used to evaluate generalization ability. Its evaluation metrics are limited to size/depth and do not involve expression recovery or learning performance .
>
> 2) **Continuous symbolic regression benchmarks** (LLM-SRBench, SRBench) operate in the real number field with continuous operators, which are completely incompatible with Boolean logic semantics; Its accuracy index relies on continuous residuals (such as NMSE, R²) [4,5]. Furthermore, the explosive combinatorial nature of Boolean logic expressions makes them significantly superior to continuous SR in terms of structural scale and solution difficulty.
>
> 3) Although **Boolean network benchmarks** (BBM and MCBF) [6,7] do contain Boolean structures, they suffer from several inherent limitations: their scales are fixed, the operator sets cannot be adjusted, and they lack support for noise modeling, sampling mechanisms, multi-task settings, and systematic complexity control.
>
> In contrast, LogicSR provides broader coverage by supporting both logic synthesis (LS) and symbolic regression (SR) tasks within a unified framework.
>
> We include the table in **Section 1** (**Table 1**) and provide a detailed comparison in **Appendix A.11**.
>
> **References**
>
> [1] Amarú, L., Gaillardon, P.-E., & De Micheli, G. (2015). The EPFL combinational benchmark suite. International Workshop on Logic Synthesis (IWLS).
>
> [2] Brglez, F., & Fujiwara, H. (1985). A neutral set of 10 combinational benchmark circuits. IEEE International Symposium on Circuits and Systems (ISCAS).
>
> [3] IWLS Programming Contest 2025. (2025). Retrieved from https://www.iwls.org/iwls2025/
>
> [4] Shojaee, P., Nguyen, N. H., Meidani, K., Farimani, A. B., Doan, K. D., & Reddy, C. K. (2025). LLM-SRBench: A new benchmark for scientific equation discovery with large language models. arXiv preprint arXiv:2504.10415.
>
> [5] La Cava, W., Orzechowski, P., Burlacu, B., et al. (2021). Contemporary symbolic regression methods and their relative performance. NeurIPS Datasets and Benchmarks.
>
> [6] Pastva, D., Safránek, S., et al. (2023). BioDivine Boolean Models (BBM) dataset. Zenodo.
>
> [7] Subbaroyan, A., Martin, O. C., & Samal, A. (2022). Minimum complexity drives regulatory logic in Boolean models of living systems. PNAS Nexus, 1(1), pgac017.

---

> ### Author Response · Authors · 2025-11-28
>
> >**W1: The paper proposes a novel set of benchmarks. However, the organization of the manuscript makes it difficult to locate and assess the key information. For example, the complexity characteristics of the generated benchmarks are not clearly presented in the main content. In addition, more insightful comparisons with other similar benchmarks are missing. Please refer to the detailed questions below.**
>
> **Response:** Thanks for the comments. We have adjusted the organization to present the key information more clearly. First, following Q1–Q4, we added **Table 1** in **Section 1** for benchmark comparison and provided a detailed discussion in **Appendix A.11**. We also introduced a dedicated noise model section (**Section 3.3**) and added the benchmark complexity indicators and corresponding data in **Section 4.1**. In addition, we fixed several typographical errors, improved the fluency of presentation, and refined other organizational issues.

---

### Official Review · Reviewer_12eU · 2025-10-30

**Soundness:** 3
**Presentation:** 2
**Contribution:** 3
**Rating:** 6
**Confidence:** 2

**Summary:**

This paper introduces LogicSR, a unified benchmark for discovering logical expressions from data. It bridges the gap between symbolic regression, which targets continuous functions, and logic synthesis, which assumes complete, noiseless specifications. LogicSR combines real-world datasets from digital circuits and biological networks with a scalable synthetic generator that creates diverse Boolean formulas under noise and incompleteness. The authors evaluate 14 methods across four paradigms (logic synthesis, symbolic regression, neural, and LLM-based). Results show that traditional logic synthesis excels on small tasks but fails to generalize, symbolic and neural methods handle larger scales at higher cost, and current LLMs struggle with complex logical reasoning. LogicSR thus establishes a rigorous foundation for cross-domain evaluation in logical discovery.

**Strengths:**

- Logical discovery is becoming crucial for interpretable and neuro-symbolic AI, yet lacks a unified benchmark.
LogicSR clearly fills this gap, making the paper’s contribution broadly valuable.

- The inclusion of both real-world and synthetic datasets, with controlled levels of noise and data incompleteness, enables evaluation under realistic conditions, which was missing in prior symbolic regression or logic synthesis work.

- The study connects traditionally isolated communities (EDA, symbolic regression, neuro-symbolic AI, LLMs).
The quantitative analysis across scales, operators, and noise levels gives an excellent overview of capability boundaries.

- The paper details generation algorithms, metrics, and reproducibility measures (open-source plan, parameter documentation), making the benchmark credible and replicable.

- The observation that LLMs (even GPT-o3) fail on complex Boolean reasoning is both surprising and informative for the broader research community.

**Weaknesses:**

- Although logical discovery is interesting in general, the paper could be strengthened by demonstrating concrete downstream use cases, such as how the discovered logical expressions could enhance interpretability in scientific modeling, improve circuit design efficiency, or support symbolic reasoning in neuro-symbolic systems. Without such examples, the broader practical impact remains abstract, and elaborated discussions are encouraged.

- The benchmark currently excludes XOR, NAND, implication, or higher-arity operators, which limits expressiveness and generalization analysis.

- The LLMs are only tested with single-prompt fitting tasks. Multi-step prompting or program-of-thought reasoning (which could improve symbolic regression) is not considered, leaving the evaluation somewhat incomplete.

- The two-stage synthetic data generator is central to the paper, but the influence of its parameters (priority decay, layer weighting, merging strategy) is not systematically analyzed.

- While some limitations can be inferred, they are not explicitly stated. An elaborated discussion on limitations or failure cases of the proposed approach would strengthen the paper.

**Questions:**

- How would LogicSR extend to non-Boolean (e.g., multi-valued or fuzzy logic) functions?

- How would LogicSR scale to more complex reasoning tasks with implications?

---

> ### Author Response · Authors · 2025-11-22
>
> We thank the reviewer for the valuable feedback. We appreciate the opportunity to address these concerns and further improve the quality of our work. Below we provide analysis and clarifications in response.
> ____________
>
> >  **W2: The benchmark currently excludes XOR, NAND, implication, or higher-arity operators, which limits expressiveness and generalization analysis.**
>
> **Response:** Thanks for the comments. We believe that the currently used set of AN/ANO (AND/NOT/OR) operators is sufficient to cover the core range of the above two points. The reasons include:
>
> **1) Expressiveness:** In Boolean algebras, XOR, NAND, and implicationcan be expressedequivalently through AND/OR/NOT, so only AN/ANO can cover the same function space. This is consistent with the AIG (AND Inverter Graph) [1,2] form widely used in modern logical synthesis and does not limit the overall expressive power.
>
> **2) Generalization:** The essence of generalization depends on whether the model can restore the structured logical pattern. The study of combinatorial logic within the AN/ANO range has exposed the generalization weaknesses of methods in key dimensions such as depth, noise, partial sampling, and structure sharing. Therefore, our experimental conclusions should be universal for other operators.
>
> Furthermore, LogicSR's two-stage generator and evaluation framework are essentially operator-agnostic and can be directly extended to any 2-ary operators.
>
> Since our benchmark is focused on Boolean Network rather than covering all operator vocabularies, extending the synthetic generator to native k-ary operators can be a direction for future work.
>
> **References**
>
> [1] Mishchenko et al., “ABC: A System for Sequential Logic Synthesis,” 2020； [2] Amaru et al., “The EPFL Combinational Benchmark Suite,” IWLS 2015
>
> _____________
>
> > **W3: The LLMs are only tested with single-prompt fitting tasks. Multi-step prompting or program-of-thought reasoning (which could improve symbolic regression) is not considered, leaving the evaluation somewhat incomplete.**
>
> **Response:** According to your suggestion, we supplemented the corresponding experiments. The following are the preliminary results of PoT reasoning [1] (in5_out5).
>
> **1) GPT-o3:**
>
> | Opereator Set | Noise      | Bit-Wise Acc | Sample-Wise Acc | Per-Output Correctness | Avg_Num_Gates | Time     |
> | ------------- | ---------- | ------------ | --------------- | ---------------------- | ------------- | -------- |
> | ANO           | noise_0    | 0.9968       | 0.9851          | 0.9707                 | 40.9993       | 285.1090  |
> |               | noise_0.01 | 0.9929       | 0.9754          | 0.9627                 | 43.5327       | 271.6047 |
> |               | noise_0.05 | 0.9896       | 0.9657          | 0.9427                 | 82.9247       | 323.5353 |
> | AN            | noise_0    | 0.9917       | 0.9686          | 0.9500                   | 44.7947       | 279.4177 |
> |               | noise_0.01 | 0.9905       | 0.9660           | 0.9473                 | 45.7820        | 259.6735 |
> |               | noise_0.05 | 0.9820        | 0.9364          | 0.9043                 | 83.7707       | 330.0995 |
>
> **2) GPT-4o:**
>
> | Opereator Set | Noise Level      | Bit-Wise Acc | Sample-Wise Acc | Per-Output Correctness | Avg_Num_Gates | Time    |
> | ------------- | ---------- | ------------ | --------------- | ---------------------- | ------------- | ------- |
> | ANO           | noise_0    | 0.6944       | 0.2495          | 0.1679                 | 21.6820        | 44.3033 |
> |               | noise_0.01 | 0.7022       | 0.2703          | 0.1827                 | 24.9027       | 35.8762 |
> |               | noise_0.05 | 0.6860        | 0.2481          | 0.1647                 | 26.3940        | 41.3840  |
> | AN            | noise_0    | 0.6955       | 0.2668          | 0.1740                  | 21.5080        | 41.9892 |
> |               | noise_0.01 | 0.7016       | 0.2690           | 0.1707                 | 23.0600         | 35.7554 |
> |               | noise_0.05 | 0.6839       | 0.2616          | 0.1760                  | 25.5173       | 39.8624 |
>
> Our primary experimental results show that incorporating PoT multi-step reasoning improves bit-level and sample-level accuracy for GPT-4o and GPT-o3, but also increases inference time and expression complexity. We further observe that some high-accuracy outputs correspond to high expression complexity. In addition, at larger scales, current LLMs still fail to generate valid expressions.
>
> **References**
>
> [1] Payoungkhamdee, P., Tuchinda, P., Baek, J., Cahyawijaya, S., Udomcharoenchaikit, C., Manakul, P., Limkonchotiwat, P., Chuangsuwanich, E. and Nutanong, S., 2025. Towards better understanding of program-of-thought reasoning in cross-lingual and multilingual environments. arXiv preprint arXiv:2502.17956.

---

> ### Author Response · Authors · 2025-11-22
>
> > **W4: The two-stage synthetic data generator is central to the paper, but the influence of its parameters (priority decay, layer weighting, merging strategy) is not systematically analyzed.**
>
> **Response:** We appreciate your suggestion and have conducted ablation experiments to analyze the influence of three key settings in our two-stage synthetic data generator. Besides truth table uniqueness, we also introduced three new metrics to evaluate network structures: 1) Fanout Layer Variance (FLV): measures the mean of node fanout variance across layers. 2) Maximum Depth: length of the longest path from inputs to any nodes in the network. 3) Normalized Depth: Scaled depth against network size (Max_Depth/$log_{2}$(Network_Size)}).
>
> In each experiment, we generated 50 networks and takes the average of the metrics, and all tests are done on the basis connectives set {AND,NOT} ({AND,NOT,OR} also tested and have the same trend).
>
> **1) Priority Decay Coefficient ($\mu$)**
>
> Priority decay coefficient $\mu$ reduces the priority of parent nodes whenever a new node is added, preventing logic reuse and yielding deeper yet structurally stable networks.
>
> **Experimental Setup:** Fixed scale (10 inputs,10 outputs) networks generated by merging fixed scale sub-network combinations.
>
> | Setting               | $\mu$ | FLV     | Max_Depth | Normalized_Depth | Global_Uniqueness |
> |-----------------------|-------|---------|-----------|-------------------|--------------------|
> | No_decay              | 1     | 0.7482  | 17.97     | 2.660              | 0.986              |
> | **Default ($p_0$)**      | 3     | 0.6214  | 19.51     | 2.894             | 0.990               |
> | Stronger decay ($p_0^2$) | 9     | 0.6271  | 21.75     | 3.236             | 0.989              |
>
>
> The experimental results show that stronger decay encourages deeper structures, while our baseline achieves optimal balance with the highest global uniqueness (0.99) and lowest structural fluctuation (FLV).
>
> **2) Layer Weight Exponent Analysis ($\alpha$)**
>
> $\alpha$ comes from the weighting function $w(v) = \bigl(\operatorname{priority}(v) + \sqrt{| \mathrm{Anc}(v) | / p_{0}}\bigr)^{\alpha}$.
>
> **Experimental Setup:** scale (20 inputs, 20 outputs, 640 gate nodes) generated by merging fixed-scale baseline configurations (two subnetworks, each with 8 inputs 8 outputs and 170 gate nodes).
>
> | Setting        | $\alpha$ | FLV    | Max_Depth | Normalized_Depth | Global_Uniqueness |
> | -------------- | ----- | ------ | --------- | ---------------- | -------------- |
> | Linear         | 1.0     | 0.4885 | 122.54    | 13.08            | 0.901          |
> | **Default**       | 1.5   | 0.4000    | 132.80     | 14.18        | 0.928   |
> | Strong Amplify | 3.0     | 0.4838 | 123.82    | 13.22     | 0.912    |
>
> Baseline ($\alpha=1.5$) provides the best uniqueness with highest depth, and FLV is the smallest. The exponential weighting ($\alpha >1$) is crucial for emphasizing deeper nodes (enhancing benchmark complexity) while maintaining diversity.
>
> **3) Merging Strategy Analysis (Number of Remaining Steps)**
>
> In this experiment, we analyze the effect of the remaining steps, defined as the difference between the target gate count $M$ (here, 640) and the total number of gate nodes already provided by the sub-networks.
>
> **Experimental Setup:**  (40 inputs, 40 outputs, 640 gate nodes) generated by subnetworks with different scales (6 input, 5 output, 6 subnetworks, each with different number of gate nodes). The number of remaining steps is the difference between the target gate number M (here 640) and the total number of gate nodes provided in the sub-networks.
>
> | Setting | Number of Remaining Steps | FLV    | Max_Depth | Normalized_Depth | Global_Uniqueness |
> | ------------------------- | ------------------------- | ------ | --------- | ---------------- | -------------- |
> | **Default (Large)**     | 480     | 0.4210  | 90.12     | 9.578     | 0.982      |
> | Medium       | 300    | 0.5236 | 73.18     | 7.777        | 0.978     |
> | Small         | 100   | 0.6562 | 51.48     | 5.471       | 0.869    |
>
> The experiment with a large number of remaining steps achieves the lowest FLV, maximum depth, and highest uniqueness. This confirms that combining partial subnet usage with more newly generated gates prevents structural repetition and maintains an adequate depth.
>
> **4) Other Parameters**
>
> Other parameters, like the setting of initial $p_0$, are also tested. However, within a reasonable range (1-12), all metrics did not change much, so results are not shown here. For merging stage, the number of remaining inputs/outputs and the number of sub-networks are also tested. Performance remains robust across all configurations and metrics.
>
> The experiments reveal three key insights: 1) Priority decay prevents structural imbalance; 2) Power-law weighting encourages depth-uniqueness tradeoff; 3) Controlled subnet reuse maintains complexity.
> We have updated these contents in **Appendix A.4.3**.

---

> ### Author Response · Authors · 2025-11-22
>
> > **W5: While some limitations can be inferred, they are not explicitly stated. An elaborated discussion on limitations or failure cases of the proposed approach would strengthen the paper.**
>
> **Response:** Thank you for this constructive suggestion. We agree discussions on limitations would enhance the presentation. The current limitations of the proposed method and directions for future work are as follows:
>
> **1) Structural Constraints:** The binary fan-in design ensures validity but limits expressiveness for highly interconnected circuits, while the simplified operator set may not fully capture optimization challenges in industrial gate libraries (e.g., AOI, MUX). Extending to multi-fanin nodes and technology-mapped operators would better reflect industrial complexity.
>
> **2) Scalability Constraints:** Evaluation remains constrained to practical-scale problems. Distributed frameworks and hierarchical generation could enable benchmarking of industrial ultra-large-scale systems (e.g., ASICs).
>
> **3**)**Domain Applications**: While the benchmark focuses on general Boolean regression, the synthetic data generation currently lacks domain-specific inductive biases; in this work, we compensate for this by incorporating real-world datasets, but future extensions may further explore domain-knowledge-driven synthetic logic generation and richer forms of logic found in real systems, such as temporal logic in control and scheduling scenarios.
>
> We will provide a detailed description in the updated version.
>
> ___________
>
> > **Q1: How would LogicSR extend to non-Boolean (e.g., multi-valued or fuzzy logic) functions?**
>
> **Response:** Thanks for the comments. LogicSR evaluates the effectiveness of logic function construction in a logic system, LogicSR can be extended to non-Boolean functions by the following measures:
>
> **1) Extended Operator Set:** Generalize binary operator sets to k-ary operators (e.g., ternary MIN/MAX) and continuous-value operators (e.g., t-norms for fuzzy logic). During the second-stage merging of the algorithm, parent node selection extends to sequential sampling without replacement of $k$ nodes, with other strategies preserved or adaptively optimized for specific settings.
>
> **2) Generalized Data Representation:** Extend the binary value set {$0,1$} to discrete sets {$0,...,K-1$}, expanding node representation vectors from $2^n$ to $K^n$ for multi-valued logic. For fuzzy or probabilistic logic, continuous sampling replaces enumeration with t-norm/t-conorm computation.
>
> **3) Evaluation Metrics:** The current bit-wise and sample-wise accuracy metrics can be naturally extended to multi-class accuracy for multi-valued logic, or to continuous similarity measures for fuzzy logic.
>
> ___________
>
> > **Q2: How would LogicSR scale to more complex reasoning tasks with implications?**
>
> **Response:** Thanks for this question. Scaling to more complex reasoning tasks with implications requires handling of relationships between logical rules, for example, $A \to B$ or chained rules $A \to B \to C$. Although these tasks fall under the category of rule-based reasoning, they can ultimately be encapsulated in a Boolean function: The current operator set is completely adequate in terms of expressive power, since $A \to B$ can be expanded using AND/NOT ($\lnot A \lor B$). The following complementary methods could make LogicSR naturally extend to such tasks:
>
> **1) Extend Operator sets to support Implication:** Implication ($A \to B$) could be directly added to the operator sets of LogicSR without any modifications, as it is a 2-ary operator and every node could be regarded as an implication relation between its parenting nodes.
>
> **2) Modelling to a higher-depth network:** As multi-step implication could be represented by network structures with a higher depth, LogicSR's two-stage synthesis framework could increase the network depth by including more gating nodes and input variables in the network structures, without introducing any new mechanisms.

---

> ### Author Response · Authors · 2025-11-28
>
> >**W1：Although logical discovery is interesting in general, the paper could be strengthened by demonstrating concrete downstream use cases, such as how the discovered logical expressions could enhance interpretability in scientific modeling, improve circuit design efficiency, or support symbolic reasoning in neuro-symbolic systems. Without such examples, the broader practical impact remains abstract, and elaborated discussions are encouraged.**
>
> **Response:** Thanks for the comments. The related task domains involved in LogicSR can all be regarded as typical application scenarios in the neuro-symbolic system:
>
> **1) Biological domain:** In biological Boolean networks (such as BBM), the regulatory relationships of genes are usually modeled using Boolean update rules to describe which regulatory factors activate or inhibit target genes under what combination conditions. Real biological data are affected by sequencing errors, cellular heterogeneity and experimental limitations, and thus can only observe sparse and noisy discrete gene states. The significance of LogicSR lies in evaluating the ability to restore logical dependencies under noise and partial observation. This ability can help propose more interpretable and analyzable regulatory hypotheses. Related works such as LogicNet [1] model gene expression as a probability distribution, and jointly identify network topology and underlying logical structure; Research [2] also systematically studied the accuracy of Threshold rules in Simple Boolean Threshold Networks relative to real biological logic.
>
> **2)  EDA field:** Under the condition that a certain error is allowed, the Boolean functions recovered through learning can be used as interpretable approximations of circuit modules. For example, the average relative error of the FPGA approximate multiplier of SMApproxLib is within 0.126, and the complexity is approximately within 64 LUTs [3]; TinyClassifier [6] can automatically generate a table data classification predictor circuit, with the number of gates controlled within 300 and an accuracy of 40-100% on 33 tasks.
>
> In addition, EDA pipelines typically span multiple stages such as logic synthesis, optimization, and mapping. The ability to directly restore Boolean functionality from data means that some costly steps can be skipped, thereby enhancing design efficiency.
>
> We have added these contents in **Appendix A.9**.
>
> **References**
>
> [1] Malekpour, S. A., Alizad-Rahvar, A. R., & Sadeghi, M. (2020). LogicNet: Probabilistic continuous logics in reconstructing gene regulatory networks. BMC Bioinformatics, 21, 318.
>
> [2] Kadelka, C., & Hari, K. (2025). Critical assessment of the ability of Boolean threshold models to describe gene regulatory network dynamics. PNAS Nexus, 4(8), pgaf228.
>
> [3] Ullah, S., Murthy, S. S., & Kumar, A. (2018). SMApproxLib: Library of FPGA-based approximate multipliers. In Proceedings of the 55th Annual Design Automation Conference (DAC) (pp. 1–6).
>
> [4] Iordanou, K., Atkinson, T., Ozer, E., Kufel, J., Aligada, G., Biggs, J., Brown, G., & Luján, M. (2024). Low-cost and efficient prediction hardware for tabular data using tiny classifier circuits. Nature Electronics, 7(5), 405–413.

---

### Official Review · Reviewer_FUaD · 2025-10-31

**Soundness:** 3
**Presentation:** 2
**Contribution:** 3
**Rating:** 6
**Confidence:** 4

**Summary:**

This work introduces **LogicSR**, a benchmark that evaluates 14 algorithms (including solvers, ML models and LLMs) on logical symbolic regression -- the task of discovering boolean expressions from data. The benchmark consists of two real-world datasets (from (i) biology and (ii) circuit) and a synthetic dataset, spanning a wide range of scalability and granularity. The 14 algorithms were evaluated using a diverse set of metrics considering scale, accuracy, robustness to noice, and efficiency. The key findings are:
- traditional logic synthesis methods perform well at small scales but lack generalization
- ml methods have better generalization ability but requires higher computation costs
- llms evaluated (gpt-o3 and gpt-4o) show limited capability

**Strengths:**

S1: the motivation was clearly identified - the gap between Symbolic Regression benchmarks and Logic Synthesis benchmarks. This benchmark gives a more comprehensive evaluation of how the input method performs given the task of finding underlying logical rules from the data.

S2: the selected methods being tested span traditional methods, ml methods and llms, which shows the compatibility of the proposed benchmark.

S3: analysis on scale, accuracy, robustness to noice, and efficiency was presented in detail, proving insightful findings for the capability of each selected method.

**Weaknesses:**

W1: it is not clear to the reader how noice level is defined in section 4.1, are the noisy data generated synthetically? Coming from real-life examples? Simple mutants of a valid data?

W2: the complexity metrics is confusing for the case of ML methods and LLMs. Since they do not output a set of logical expressions, how was complexity computed (the definition of complexity was to count the number of operators, are llms prompted to output the logical expressions?)?

W3: inconsistency/confusion in cost measurement. It was mentioned that for non-ml methods the efficiency metric refers to the total processing time, and for ml methods it refers to model training time. However, in section 4.4 training time for ml methods were not discussed. In addition, no cost/efficiency discussion is included for LLMs. Were they fine-tuned? API-called? Evaluated by inference time?

W4: some writing/presentation issues will be mentioned in the question section.

**Questions:**

Q1: line 078-085 seem to contain repetitive contents and poorly written.

Q2: line 164 - is $M_j$ a fixed value? how is this value chosen?

Q3: Why are input and output always the same for synthetic data (in table 1)? any justification for that choice? could asymmetric input-output values significantly impact the performance?

Q4: Figure 5 what happens to GPT-4o and GPT-o3?

Q5: any consideration of including SAT/SMT learning methods (seem to be related as their tasks are also to learn a set of underlying boolean expressions from data, often in CNF/DNF forms). E.g. [1]

[1] Wang, Po-Wei, et al. "Satnet: Bridging deep learning and logical reasoning using a differentiable satisfiability solver." International Conference on Machine Learning. PMLR, 2019.

---

> ### Author Response · Authors · 2025-11-22
>
> We thank the reviewer for the valuable feedback. We appreciate the opportunity to address these concerns and further improve the quality of our work. Below we provide analysis and clarifications in response.
>
> _____________________________________
>
> > **W1: it is not clear to the reader how noise level is defined in section 4.1, are the noisy data generated synthetically? Coming from real-life examples? Simple mutants of a valid data?**
>
> **Response:** Thank you for this question. In LogicSR, all noisy datasets are synthetically generated to ensure precise noise control, rather than created through mutants or extracted from real-life examples.
>
> Noise is introduced by performing deterministic bit-flips on the outputs' truth tables. Specifically, for each Boolean function with $K$ inputs, we randomly select exactly $\lfloor \eta \cdot 2^K \rfloor$ input combinations for each output variable (where $\eta \in$ {$0, 0.01, 0.05$} is the noise level), and flip corresponding output bits in the truth table.
>
> Bit-flip perturbations simulate the systematic errors commonly observed in biological measurements and circuit-level sampling, and are widely used as a standard noise model in these domains [1–3]. In our experiments, we use the fixed-budget noise model [4], which allows the noise magnitude at each preset level to be controlled in a precise and reproducible manner.
>
> **References**
>
> [1] Trajanovski, S., Martin-Hernandez, J., Winterbach, W., & Van Mieghem, P. (2015). Modeling and inference of noisy Boolean dynamics.
>
> [2] Baumann, R. C. (2005). Radiation-induced soft errors in advanced semiconductor technologies. IEEE Transactions on Device and Materials Reliability.
>
> [3] Krishnan, H. (2011). Modeling and simulation of transient faults in logic circuits. IEEE Transactions on Circuits and Systems.
>
> [4] Natarajan, N., Dhillon, I. S., Ravikumar, P., & Tewari, A. (2013). Learning with noisy labels. Advances in Neural Information Processing Systems (NeurIPS 2013), 26, 1196–1204.
>
> _____________________________________
>
> > **W2: the complexity metrics is confusing for the case of ML methods and LLMs. Since they do not output a set of logical expressions, how was complexity computed (the definition of complexity was to count the number of operators, are llms prompted to output the logical expressions?)?**
>
> **Response:** Thanks for your comments. First of all, it should be clarified that LLMs are capable of outputting grammatically valid expressions according to the rules.
>
> For  Difflogic, Neuralut, and AI4EDA_TNet in ML methods  (Boolformer can generate expressions)，they only output the predicted truth table. However, due to the certain correspondence between their network architecture and Boolean expressions, We can infer the complexity of expressions from the construction semantics of the network. As detailed below:
>
> **1) Difflogic:** The DiffLogic stack $L(k)$ layers of differentiable Boolean units, each layer containing $h(k) $ soft and OR gates; The first layer $L(k)-1$ provides an intermediate connection/disjunction structure, while the last layer collapses into an equivalent lut enumeration of all output configurations, resulting in $ [L (k)-1]\cdot h(k)+2^k$.
>
> **2) Neuralut:** NeuraLUT parameterizes a lookup table with $2^k$ entries for $k$ outputs, which corresponds to a canonical implementation with $2^k$ minimum items; Therefore, its Boolean equivalent circuit complexity is $2^k $.
>
> **3) AI4EDA-TNet:** AI4EDA-TNet adds a LuT-like output layer with $h$ hidden gate units, each corresponding to a logical operation, resulting in $h+2^k$.
> _______________________________

---

> ### Author Response · Authors · 2025-11-22
>
> > **W3: inconsistency/confusion in cost measurement. It was mentioned that for non-ml methods the efficiency metric refers to the total processing time, and for ml methods it refers to model training time. However, in section 4.4 training time for ml methods were not discussed. In addition, no cost/efficiency discussion is included for LLMs. Were they fine-tuned? API-called? Evaluated by inference time?**
>
> **Response:** We agree that a unified definition is necessary. Across all categories of methods, efficiency refers to the wall-clock time required to produce the final expressions (or predictions) for a given dataset instance. This definition is implementation-agnostic and applies uniformly to logic-synthesis, symbolic, neural, and LLM-based approaches.
>
>  **1) Logic-synthesis / symbolic-regression (non-learning) methods:** The reported cost corresponds to the *single-pass solving time*, i.e., loading the dataset, generating expressions from the training set, and producing predictions on the test set using the generated expressions.
>
> **2) Machine-learning-based methods:** For DiffLogic, AI4EDA-TNet, NeuraLUT, the reported cost is the *end-to-end time*, including both training and inference. This consists of training the neural network on the training set and then predicting on the test set using the trained model. For Boolformer, since a pretrained model is used, we report only the time to generate expressions from the pretrained model and the inference time on the test set.
>
> **3) LLMs**: No fine-tuning is performed. The total time includes the API calls used to generate expressions from the training set.
>
> We have included this explanation in **Appendix A.7.3**.
>
> _____________________________________
>
> > **Q2: line 164 - is $M_j$ a fixed value? how is this value chosen?**
>
> **Response:** In LogicSR, $M_j$ is not a fixed constant but a configurable hyperparameter used to specify the number of internal logic gates to be generated in the subnetwork. In actual settings, In practice, the value of $M_j$ is chosen to match the desired target complexity, while respecting the overall scaling constraints, including the global depth, the total gate budget, and the feasibility of subsequent subnetwork merging.
>
> _____________________________________
>
> > **Q3: Why are input and output always the same for synthetic data (in table 1)? any justification for that choice? could asymmetric input-output values significantly impact the performance?**
>
> **Response:** Thank you for this question. We set input equal to output in synthetic data primarily to ensure: 1) consistent scaling across difficulty levels, 2) comparable multi-output structures, and 3) controlled complexity for generalization evaluation. This symmetric design helps isolate the effect of input dimensionality, which is the deciding factor of single logic formula complexity.
>
> However, this is not a limitation of our generator, which fully supports asymmetric configurations ($K \neq L$). Real-world datasets (IWLS and BBM) already include various asymmetric cases in our evaluation. Therefore, maintain interpretability and complement the asymmetric configurations present in real-world datasets, we designed the synthetic data to be symmetric.
>
> Experiments on generating AN networks show that symmetric designs exhibit high structural stability. They achieve relatively low Fanout Layer Variance (FLV, 0.32–0.53), indicating more uniform connectivity, maintain reasonable maximum depth, produce well-scaled normalized depth, and yield high output uniqueness (>0.85).
> In contrast, asymmetric configurations display noticeably higher FLV (0.28–0.84) and reduced uniqueness (0.62–0.74). This behavior is expected, as the limited number of distinct input patterns inherently caps the achievable uniqueness when the input dimension is smaller than the output dimension. Further explanation is provided in **Appendix A.4.3**.
>
> | In_Size | Out_Size | Avg_Uniqueness | Max_Depth | Normalized_Depth | Avg_FLV |
> | ------- | -------- | -------------- | --------- | ---------------- | ------- |
> | 5       | 5    | 0.8573     | 10.27     | 2.11   | 0.3181  |
> |    | 10    | 0.7421  | 8.87      | 1.84    | 0.4528  |
> | 10      | 5   | 0.6261         | 86.71     | 10.24     | 0.4214  |
> |    | 10  | 0.9938         | 47.46   | 5.66    | 0.6335  |
> |    | 20       | 0.9903      | 31.37     | 3.80   | 0.8390   |
> | 20      | 10       | 0.6400      | 230.80     | 24.64    | 0.2811  |
> |      | 20   | 0.9200   | 132.84    | 14.18     | 0.3986  |
> |    | 30       | 0.9584      | 102.18    | 10.91    | 0.4904  |
> | 40      | 20       | 0.6327   | 105.60     | 11.22    | 0.3766  |
> |    | 40  | 0.8778      | 61.41     | 6.53       | 0.5267  |
> |      | 80       | 0.8680     | 43.96     | 4.67      | 0.7255  |
>
> Therefore, the superior stability of symmetric designs makes it a preferred topology for constructing high-quality benchmarks, despite the framework's full support for both symmetric and asymmetric configurations.

---

> ### Author Response · Authors · 2025-11-22
>
> > **Q4: Figure 5 what happens to GPT-4o and GPT-o3?**
>
> **Response:** Thank you for the correction. Upon re-examining the experimental logs for Figure 5, we found that one set of GPT-4o measurements in the paper was not properly entered into the statistical table. In addition, due to context-length limitations, no valid outputs are returned when the scale becomes large, as reflected by the missing bars for input size greater than 10 in the figure.
> The following results are obtained from the re-run experiments, and we have updated **Figure 5** in **Section 4.4** accordingly.
> | Method | Settings | Noise_0  | Noise_0.01 | Noise_0.05 |
> |--------|----------|----------|-------------|-------------|
> | GPT-4o | ANO | 2.420   | 2.428    | 2.581    |
> |          | AN  | 2.505   | 2.617    | 2.861     |
> | GPT-o3 | ANO | 160.155 | 149.048  | 232.571  |
> |          | AN  | 153.021 | 144.911   | 216.989  |
>
>
> > **Q5: any consideration of including SAT/SMT learning methods (seem to be related as their tasks are also to learn a set of underlying boolean expressions from data, often in CNF/DNF forms).**
>
> **Response:** Thanks for your comments. we have not included the SAT/SMT methods because: 1) The output of such methods is usually not an explicit Boolean expression or a complete truth table. To incorporate it into LogicSR, additional expression reconstruction or CNF/DNF parsing of latent constraints is required. Furthermore, the idea of differentiable logic reasoning in SATNet [1] is similar to the Difflogic method incorporated in LogicSR.
>
> However, we recognize its value as an independent method family and include supplementary experiments. The results are summarized as follows (the value on the left of “/” denotes the training accuracy, while the value on the right denotes the test accuracy):
>
> **1) AN:**
>
> | Method | Scale       | Noise Level | Bit-Wise Acc      | Sample-Wise Acc     | Per-Output Correctness | Time     |
> |--------|-------------|-------------|--------------------|----------------------|-------------------------|----------|
> | SATNet | in5_out5    | 0           | 1                  | 1                    | 1                       | 1.262    |
> |        |             | 0.01        | 1                  | 1                    | 1                       | 1.271    |
> |        |             | 0.05        | 0.999              | 0.997                | 0.979                   | 1.250    |
> |        | in10_out10  | 0           | 0.999 / 0.996      | 0.989 / 0.969        | 0.781 / 0.727           | 12.587   |
> |        |             | 0.01        | 0.988 / 0.983      | 0.893 / 0.856        | 0 / 0.025               | 12.381   |
> |        |             | 0.05        | 0.938 / 0.925      | 0.533 / 0.467        | 0 / 0                   | 12.960   |
> |        | in20_out20  | 0           | 0.999 / 0.999      | 0.991 / 0.990        | 0.727 / 0.739           | 275.689  |
>
> **2) ANO:**
>
> | Method | Scale       | Noise Level | Bit-Wise Acc     | Sample-Wise Acc    | Per-Output Correctness | Time     |
> |--------|-------------|-------------|-------------------|----------------------|-------------------------|----------|
> | SATNet | in5_out5    | 0           | 1                 | 1                    | 1                       | 1.286    |
> |        |             | 0.01        | 1                 | 1                    | 1                       | 1.273    |
> |        |             | 0.05        | 0.999             | 0.997                | 0.978                   | 1.266    |
> |        | in10_out10  | 0           | 0.998 / 0.995     | 0.983 / 0.962        | 0.715 / 0.669           | 12.591   |
> |        |             | 0.01        | 0.988 / 0.983     | 0.891 / 0.850        | 0 / 0.031               | 12.494   |
> |        |             | 0.05        | 0.938 / 0.924     | 0.532 / 0.465        | 0 / 0                   | 12.902   |
> |        | in20_out20  | 0           | 0.999 / 0.999     | 0.990 / 0.989        | 0.732 / 0.744           | 268.019  |
>
> From the experimental results, SATNet achieves near-perfect bit-wise accuracy in noise-free settings. Under moderate noise, it can still reliably recover the structure of small-scale tasks (e.g., in5). However, when the input size increases to 10 or the noise level becomes high, both sample-wise accuracy and per-output correctness decline sharply, while the runtime rises from approximately 12 seconds to over 270 seconds. In addition, SATNet does not generate explicit symbolic expressions, which limits its interpretability and prevents direct recovery of human-readable formulas.
>
> **References**
>
> [1] Wang, P.-W., Rao, N., Fagan, J., & Kolter, J. Z. (2019). SatNet: Bridging deep learning and logical reasoning using a differentiable satisfiability solver. International Conference on Machine Learning (ICML), PMLR.

---

> > ### Author Response · Authors · 2025-11-28
> >
> > >**W4: some writing/presentation issues will be mentioned in the question section.**
> >
> > **Response:** Thanks for pointing out the problems. We have done revision based on the questions, while also fix some typographical errors, adjusting presentation fluency, and refining some organization problems.
> >
> > >**Q1: line 078-085 seem to contain repetitive contents and poorly written.**
> >
> > **Response:** The revised text is as follows:
> >
> > *Using our benchmark, we conduct a fine-grained evaluation of 17 representative algorithms, covering classical logic-based methods, modern machine-learning models, and large language models. Overall, the results show that existing approaches struggle with moderately scaled logical discovery tasks, either requiring substantial computational resources or delivering unsatisfactory accuracy. More specifically:
> > (1) traditional logic synthesis performs well on small-scale problems but lacks scalability and flexibility;
> > (2) ML and symbolic-regression models generalize better but come with significant computational overhead; and
> > (3) current LLMs still show limited capacity for handling complex logical reasoning.
> > These findings collectively reveal challenges in scalability, noise robustness, and operator-set compatibility across current methods.*
> >
> > This revised text has also been incorporated into **Section 1**.

---

### Author Response · Authors · 2025-11-22
**The Overall Response to the Reviewers**

We sincerely appreciate the reviewers for their detailed and constructive feedback on our submitted manuscripts. We are glad that they have recognized the originality and importance of our work. Our benchmark LogicSR is designed to offer structured challenges for logic synthesis and symbolic regression. To the best of our knowledge, LogicSR is the first evaluation benchmark for symbolic regression tasks that simultaneously supports Boolean operators, controllable noise, partial observability, multi-output settings, and scalable deep logic structures.

Several recent studies in logical reasoning, symbol discovery, and neuro-symbolic methods (such as the SRBench series [2,3], IWLS [1], and BBM [4]) mainly rely on traditional logical synthesis datasets or continuous symbolic regression datasets in their evaluation systems, and have not fully integrated cross-domain characteristics and methods. In contrast, LogicSR has advantages in method compatibility and discussion of the capability boundaries of existing methods, thereby filling the gap in existing benchmarks.

Although Boolean logic is crucial in fields such as circuit design and neuro-symbolic reasoning, most current evaluation methods emphasize exact fitting and cannot be used for learning and generalization. We believe that LogicSR offers a unified and scalable starting point, which combines real Boolean networks, controllable synthetic logical structures, and generation-oriented learning tasks, enabling researchers to make fair comparisons on common tasks across domains and methods. We look forward to the feedback from the community on our contributions.

**References**

[1] IWLS Programming Contest 2025. Available at: https://www.iwls.org/iwls2025/

[2] Shojaee, P., Nguyen, N. H., Meidani, K., Farimani, A. B., Doan, K. D., & Reddy, C. K. (2025). LLM-SRBench: A new benchmark for scientific equation discovery with large language models. arXiv preprint arXiv:2504.10415.

[3] La Cava, W., et al. (2021). Contemporary symbolic regression methods and their relative performance. NeurIPS Datasets and Benchmarks.

[4] Pastva, D., Safránek, S., et al. (2023). BioDivine Boolean Models (BBM) Dataset. Zenodo.

---

### Note · Program_Chairs · 2026-01-17
**Submission Desk Rejected by Program Chairs**

The following references in this submission do not refer to real documents and/or have major errors in bibliographic information:


Hari Krishnan. Modeling and simulation of transient faults in logic circuits. IEEE Transactions on Dependable and Secure Computing, 8(4):548-558, 2011.
Stojan Trajanovski et al. Modeling and inference of noisy boolean dynamics, 2015. Preprint.